# 53BP1 regulates heterochromatin through liquid phase separation

Lei Zhang [1,6,8✉], Xinran Geng [1,8], Fangfang Wang[2,8], Jinshan Tang[2], Yu Ichida[1], Arishya Sharma[1], Sora Jin[1], Mingyue Chen[6], Mingliang Tang[3], Franklin Mayca Pozo [1], Wenxiu Wang[6], Janet Wang[4], Michal Wozniak [5,7], Xiaoxia Guo[6], Masaru Miyagi[1], Fulai Jin [4], Yongjie Xu[5], Xinsheng Yao[2] & Youwei Zhang [1✉]

Human 53BP1 is primarily known as a key player in regulating DNA double strand break (DSB) repair choice; however, its involvement in other biological process is less well understood. Here, we report a previously uncharacterized function of 53BP1 at heterochromatin, where it undergoes liquid-liquid phase separation (LLPS) with the heterochromatin protein HP1α in a mutually dependent manner. Deletion of 53BP1 results in a reduction in heterochromatin centers and the de-repression of heterochromatic tandem repetitive DNA. We identify domains and residues of 53BP1 required for its LLPS, which overlap with, but are distinct from, those involved in DSB repair. Further, 53BP1 mutants deficient in DSB repair, but proficient in LLPS, rescue heterochromatin de-repression and protect cells from stress-induced DNA damage and senescence. Our study suggests that in addition to DSB repair modulation, 53BP1 contributes to the maintenance of heterochromatin integrity and genome stability through LLPS.

[1] Department of Pharmacology, Case Comprehensive Cancer Center, Case Western Reserve University, School of Medicine, Cleveland, OH 44106, USA. [2] Institute of Traditional Chinese Medicine and Natural Products, College of Pharmacy, Jinan University, Guangzhou 510632, China. [3] College of Life Sciences, Wuhan University, Wuhan, Hubei 430068, China. [4] Department of Genetics and Genome Sciences, Case Western Reserve University, School of Medicine, Cleveland, OH 44106, USA. [5] Department of Pharmacology and Toxicology, Wright State University, Dayton, OH 45435, USA. [6] Present address: National 111 Center for Cellular Regulation and Molecular Pharmaceutics, Key Laboratory of Fermentation Engineering, Hubei University of Technology, Wuhan, Hubei 430068, China. [7] Present address: Department of Molecular Biology of Cancer, Medical University of Lodz, 6/8 Mazowiecka Street, 92-215 Lodz, Poland. [8] These authors contributed equally: Lei Zhang, Xinran Geng, Fangfang Wang. ✉email: zhanglei0222@163.com; yxz169@case.edu

53BP1 plays a critical role in double strand break (DSB) repair by promoting non-homologous end joining (NHEJ) while inhibiting homologous recombination (HR)[1,2]. 53BP1 contains an OD (oligomerization domain), two tandem Tudor domains, an UDR (ubiquitin-dependent region), an NLS (nuclear localization signal), and two BRCT (BRCA1 C-terminal repeat) domains. The formation of 53BP1 foci at DNA damage site is key for its function in DSB repair, which involves the OD-Tudor-UDR-NLS region[1,2]. While the Tudor and the UDR recognize methylated H4K20[3-6] and ubiquitinated H2AK15[7], respectively, the OD is thought to promote dimerization during DSB foci formation. The BRCTs are involved in DSB repair only in the absence of MDC1 through binding to γH2AX[8,9]. ATM-dependent phosphorylation on multiple Ser/Thr residues at the N-terminus of 53BP1 is not required for its foci formation[10]; yet, it is crucial for recruiting 53BP1's downstream factors including RIF1, PTIP, and the Shieldin complex to damage sites to promote the NHEJ pathway for DSB repair[11-18].

Heterochromatin, which can be categorized into constitutive and facultative types, refers to highly compacted DNA structures that are generally gene poor and transcriptionally silent[19,20]. While facultative heterochromatin can be regulable by cell fate[19], constitutive heterochromatin is often permanent and forms at pericentromeric, centromeric and telomeric regions that include tandem repetitive DNA sequences[21] and/or transposable elements[22,23]. Increasing evidence suggests that heterochromatin regulates various genome biology ranging from maintaining DNA structure and controlling chromosome segregation to regulating epigenetic inheritance and mediating DNA replication, repair, and transcription[19,20]. Hence, defects in heterochromatin often result in the loss of both the DNA structural integrity and the genome function, contributing to genomic instability, cellular senescence, and eventually disorders like premature ageing[24-26]. Constitutive heterochromatin commonly includes Lys 9-trimethylated histone H3 (H3K9me3), which is mainly catalyzed by SUV39H1[27,28] and recognized by the heterochromatin protein 1α (HP1α)[29,30].

Liquid-liquid phase separation (LLPS) in biology defines the process of forming membraneless liquid droplets by proteins and often nucleotides (e.g., RNA) when their concentrations have reached a threshold to allow them to phase separate from the surrounding solution[31]. Recent studies in flies, yeast, and humans showed that, when conditions were met (such as with protein–protein interaction and/or post-translational modification), core heterochromatin proteins including HP1α, SUV39H1, and TRIM28 can undergo LLPS[32-35]. However, how LLPS regulates heterochromatin and what are other important factors involved in this process remain poorly understood. Here, we report a DSB repair-independent role for 53BP1 in maintaining both the structural integrity and the transcriptional repression of heterochromatin through LLPS.

## Results

**53BP1 forms DSB foci-distinct nuclear puncta.** To explore biological functions of 53BP1, we utilized four pairs of parental and 53BP1 knockout (KO) cell lines: MDA-MB-231, U-2 OS, and MCF-10A edited by CRISPR/Cas and MEF from *53bp1* KO mice. Deletion of 53BP1 was confirmed by immunoblotting (Fig. 1a and Supplementary Fig. 1a–c) and immunostaining (Fig. 1b and Supplementary Figs. 2–5). The topoisomerase 1 poison, camptothecin (CPT), induced a similar level of CHK1 phosphorylation in parental and KO cells (Fig. 1a and Supplementary Fig. 1d–f), indicating that 53BP1 KO cells retained normal DNA damage response, consistent with the idea that 53BP1 is not a strong DNA

damage response mediator[36]. When analyzing cellular localization of 53BP1, we found that 53BP1 formed nuclear puncta in MDA-MB-231 cells under normal growth conditions, which were not observed in 53BP1 KO cells (Fig. 1b), confirming that the nuclear puncta were indeed 53BP1 signals in the absence of DNA damage. The median size of 53BP1 puncta was estimated to be 1.243 μm² (area) with the 25% and 95% percentiles to be 0.680 and 5.178 μm², respectively (Supplementary Fig. 1g). 53BP1 nuclear bodies formed under normal growth conditions have been previously reported in a small percentage of U-2 OS cells and human BJ fibroblasts as spontaneous foci, which marked damaged DNA from the previous round of the cell division cycle and were mainly presented as 1–3 large bright dots in cells[37,38] (Fig. 1b, circle). However, we observed a significantly higher number (a range of 1–40 puncta with a medium of 8 per cell) and percentage (~40% cells) of 53BP1 puncta (Fig. 1c, d) than the previously reported spontaneous foci[37,38], which led us to further investigate the nature of these 53BP1 puncta.

It was reported that 53BP1 spontaneous foci were mainly presented in G phase cells and will be resolved during late G1 to S phase through DSB repair[37,38]. Hence, we first examined the cell cycle dependency of 53BP1 puncta. We noticed that cells with 1–3 large bright dots were largely Cyclin A negative (Fig. 1e), indicating that they were at G1 phase, which is consistent with previous reports[37,38]. However, we observed an overall positive correlation between the number of 53BP1 puncta and the expression level of Cyclin A (Fig. 1e, f), suggesting that 53BP1 puncta could form during S to G2 phases of the cell cycle. These results also indicate that not all 53BP1 puncta formed under normal growth conditions represent spontaneous DNA damage foci.

Similarly, we observed 53BP1 puncta in ~30% of U-2 OS (Supplementary Fig. 2a, b), ~32% of MCF-10A (Supplementary Fig. 3a, b) and ~20% of MEF (Supplementary Fig. 4a, b) cells with a median number at around 6 per cell except in MEF (4 per cell) (Supplementary Figs. 2c, 3 and 4c). As shown in Supplementary Fig. 4f, 53BP1 puncta in MEFs are less circular in shape and under many circumstances were displayed as large irregular puncta, which resulted in, at least partially, a lower median number of 53BP1 puncta in MEF cells. Importantly, 53BP1 puncta in these cell lines positively correlated with Cyclin A expression levels and were abolished by 53BP1 KO (Supplementary Figs. 2–4). In addition, 53BP1 puncta (a median of 7 per cell) were observed in ~40% of HEK293T cells, positively correlated with Cyclin A levels, and were abolished by 53BP1 knockdown (KD) (Supplementary Fig. 5a–c). Further, 53BP1 puncta were observed in non-transformed human primary cell lines, including ~42% of lung fibroblast IMR-90 (Supplementary Fig. 6) and ~20% of retinal pigmental epithelial ARPE-19 cells (Supplementary Fig. 7). The median puncta number was ~6-7 and positively correlated with Cyclin A levels in IMR-90 and ARPE-19 cells (Supplementary Figs. 6c, d and 7c, d). These results confirm that 53BP1 puncta formation is a general phenomenon.

To further understand whether 53BP1 puncta represented spontaneous DNA damage, we analyzed foci formation of γH2AX and pKAP1, key factors involved in the DNA damage response and DSB repair, in cells showing 53BP1 puncta. Under normal growth conditions, 53BP1 puncta-positive U-2 OS (Supplementary Fig. 8a), MEF (Supplementary Fig. 8b, d) or MDA-MB-231 cells (Supplementary Fig. 8c) did not show an increase in foci formation of γH2AX or pKAP1 compared with 53BP1 puncta-negative cells. In contrast, DNA damage (CPT or bleomycin-BLEO) greatly induced foci formation of γH2AX and pKAP1 in parental and 53BP1 KO cells (Supplementary Fig. 8). These results reinforce the idea that not all 53BP1 puncta under normal growth conditions represent spontaneous damage

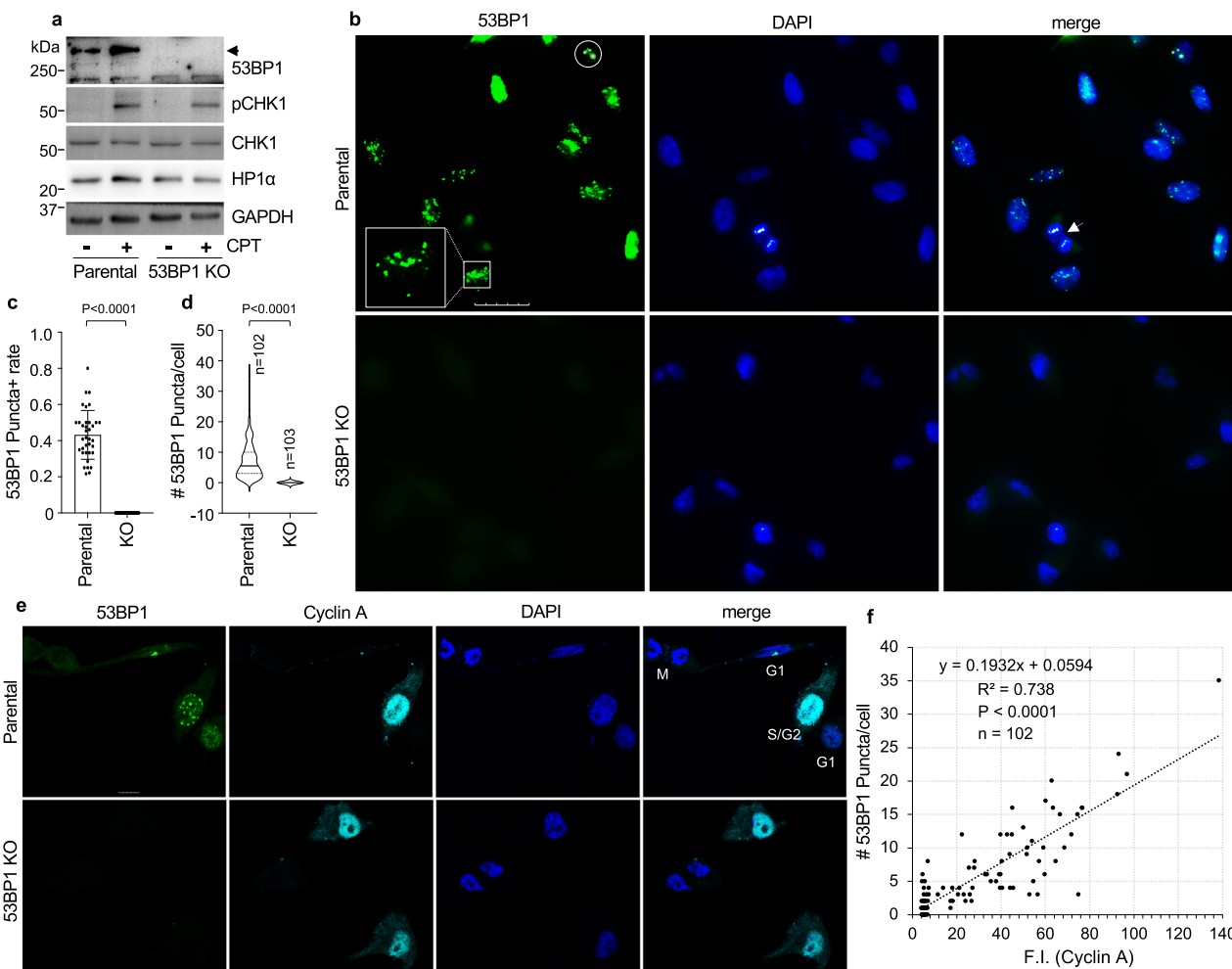

**Fig. 1 53BP1 forms nuclear puncta. a** MDA-MB-231 parental and 53BP1 knockout (KO) cells were treated or not with 500 nM CPT for 6 h, and protein expression was detected. Arrowhead indicates 53BP1. **b** Representative images of nuclear localization of 53BP1 in MDA-MB-231 cells under normal growth conditions. Single z-plane images were acquired by sequential scanning using confocal microscopy. *Square*: an example of puncta+ cell; Circle: a cell with three spontaneous 53BP1 foci; Arrow: a mitotic cell. Scale bar is 16 μm. **c** Percentage of puncta positive cells were analyzed from n = 36 individual images taken from four replicate experiments with a total n = 447 and 373 cells analyzed for parental and KO groups, respectively. Data represent mean values and standard deviation (SD). **d** Violin plot of 53BP1 puncta number per cell was analyzed from n = 185 and 105 parental and 53BP1 KO MDA-MB-231 cells, respectively. Data represent mean, 25th, and 75th percentiles with the whiskers extending to the minimum and maximum values. **e** Representative images of 53BP1 puncta and Cyclin A in MDA-MB-231 parental and 53BP1 KO cells under normal growth conditions. The cell cycle stages (G1, S/G2, and M) were determined by a combination of Cyclin A expression levels and DAPI staining patterns. Single z-plane images were acquired by sequential scanning using confocal microscopy. Scale bar is 10 μm. **f** Correlation between Cyclin A expression levels (expressed as mean fluorescence intensity, F.I.) and 53BP1 puncta number from n = 102 parental MDA-MB-231 cells. Each dot represents one cell. A size at or above 0.680 μm$^2$ (area) (≥25% percentile) was considered to be a 53BP1 punctum in the quantitative analyses. Unpaired two-tailed t test using Prism 9.0 was conducted for **c** and **d** with 95% confidence intervals, whereas the P-value in (**f**) was acquired by the Pearson Correlation Coefficient Calculator.

foci, indicating a previously uncharacterized function of 53BP1. Also, to preclude the possibility that these puncta were caused by cell culture stress, we stained U-2 OS cells with cleaved Caspase 3 (cCasp3), a marker of apoptosis. No increase in cCasp3 was observed in 53BP1 puncta positive cells, whereas treatment with BLEO significantly increased the signal intensity of cCasp3 (Supplementary Fig. 4e). These results suggest that 53BP1 puncta did not arise from cell culture-induced cell death.

**53BP1 puncta localize at heterochromatin and depend on HP1α.** To further understand these 53BP1 puncta, we asked if they co-localize with any known nuclear structures. The results show that 53BP1 puncta did not localize at SC-35-defined transcriptionally active nuclear speckles (Fig. 2a). However, we

observed a great percentage of 53BP1 puncta wrapping around DAPI-indicated heterochromatin centers in MEFs (Supplementary Fig. 4f), indicating that 53BP1 puncta were formed at heterochromatin. To test this idea, we examined the co-localization between 53BP1 and H3K9me3, a marker for constitutive heterochromatin[27]. About 44 ± 14% (mean ± SD) of 53BP1 puncta abutted or co-localized with H3K9me3 in MDA-MB-231 (Fig. 2b, c) and U-2 OS cells (Fig. 2d, e). Interestingly, we observed that the median size of H3K9me3 puncta was reduced from ~1.152 μm$^2$ in parental cells to ~0.682 μm$^2$ in 53BP1 KO cells (Supplementary Fig. 4g), which is associated with a reduction in the number of large bright H3K9me3 puncta (the top 25% percentile in size) in 53BP1 KO cells (Fig. 2f). Again, 53BP1 puncta localized to, and often wrapped around, H3K9me3-coated heterochromatin centers in MEFs (Fig. 2g). Like in human cells, *53bp1* KO resulted in a

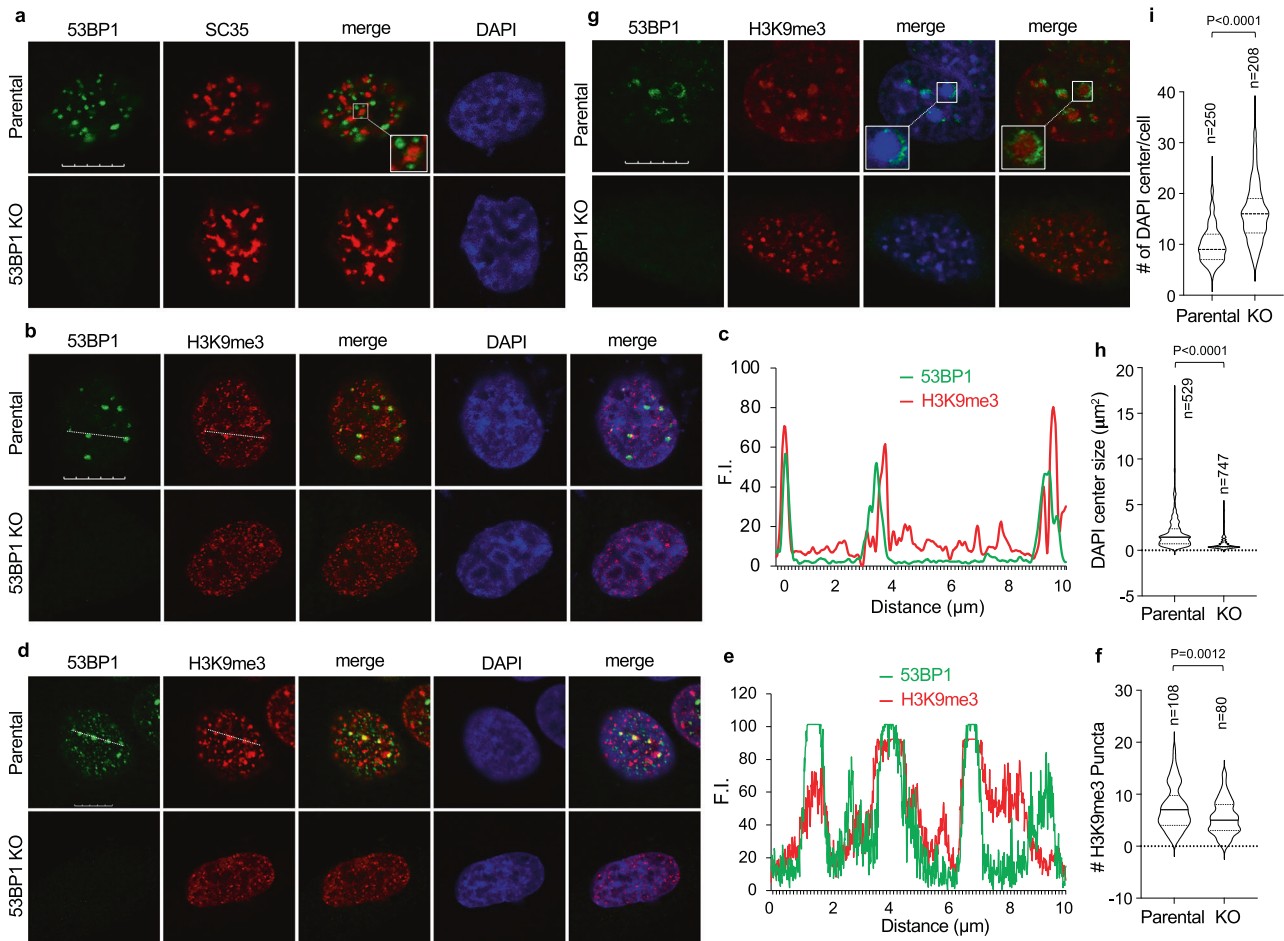

**Fig. 2 53BP1 puncta localize at heterochromatin.** Representative single z-plane confocal images from MDA-MB-231 (**a**, **b**), U-2 OS (**d**) and MEF (**g**) cells. Squares: enlarged areas. Dotted line scanning was shown for MDA-MB-231 (**c**) and U-2 OS (**e**) cells. **f** Violin plot of the number of H3K9me3 puncta (at or above the 25% percentile size of 0.746 vs 0.438 μm$^2$ for parental and KO cells, respectively) in MDA-MB-231 cells from the indicated number of cells acquired from two independent experiments. Data represent mean, 25th, and 75th percentiles with the whiskers extending to the minimum and maximum values. **h** Violin plot of the size of MEF heterochromatin centers shown by DAPI staining from indicated number of puncta acquired from two independent experiments. The size of the median, 25%, 75% and 95% percentiles of heterochromatin centers in parental and *53bp1* KO MEFs were 1.450 vs 0.408, 0.725 vs 0.362, 2.356 vs 0.906 and 5.867 vs 2.174 μm$^2$, respectively. **i** Violin plot of the number of DAPI-indicated heterochromatin centers from indicated number of cells in (**g**) acquired from two independent experiments. Data represent mean, 25th and 75th percentiles with the whiskers extending to the minimum and maximum values. Unpaired two-tailed *t* test using Prism 9.0 was performed for **f**, **h,** and **i** with 95% confidence intervals.

reduction in the size of heterochromatin centers from 1.450 in parental cells to 0.408 μm$^2$ in MEFs (Fig. 2h); however, the reduction in size was accompanied with an increase in the total number of heterochromatin in *53bp1* KO MEFs (large and small heterochromatin centers combined) (Fig. 2i).

Similarly, 53BP1 puncta partially (37 ± 13%) colocalized with HP1α, another well-known heterochromatin marker, in MDA-MB-231, U-2 OS, and MEF cells (Supplementary Fig. 9a–c). 53BP1 puncta co-localized less with HP1β (11 ± 10%, Supplementary Fig. S9d) or HP1γ (8 ± 10%, Supplementary Fig. 9e). Interestingly, in limited events when 53BP1, HP1α, and HP1γ were observed in the same puncta, HP1α localized more closely to 53BP1 than to HP1γ (Supplementary Fig. 9e). Consistently, 53BP1 co-immunoprecipitated (co-IPed) with HP1α, and to a much lesser degree HP1β, but not HP1γ in cells (Supplementary Fig. 10a). We did not detect an interaction between 53BP1 and a known binding protein, P53; however, we detected other known 53BP1-associating factors including RIF1 and MDC1 in the co-IP (Supplementary Fig. 10a). UHRF1 was also detected as a 53BP1-associating protein, consistent with the idea that UHRF1 is part of the HP1α-defined heterochromatin[39].

Using publicly available ChIP-seq data[40] collected from U-2 OS cells, we were able to compare genome-wide distribution of 53BP1 with H3K9me3. A Heat map shows a genome-wide ChIP–seq enrichment (log2[ChIP/input]) of 53BP1 and H3K9me3 on ~23% of 53BP1 peaks (Supplementary Fig. 10b), supporting the idea that 53BP1 localizes at heterochromatin. To further confirm this finding, we analyzed 53BP1 distribution on specific chromosomes[40,41]. The results show that indeed 53BP1 displayed similar binding patterns as the heterochromatin marker H3K9me3, but not the euchromatin marker H3K4me2/3, at specific locations on several chromosomes (Supplementary Fig. 10c–g), which is consistent with the immunofluorescence results where 53BP1 puncta co-localized at certain heterochromatin centers. Hence, these results support a model in which 53BP1 puncta are formed at constitutive heterochromatic regions under normal growth conditions.

We then asked if 53BP1 puncta formation depends on heterochromatin. To this end, we stably depleted HP1α, HP1β, or HP1γ in MDA-MB-231 cells, which did not affect the cell cycle or induce cell death (Fig. 3). While depletion of HP1α significantly reduced both the rate and the number of 53BP1

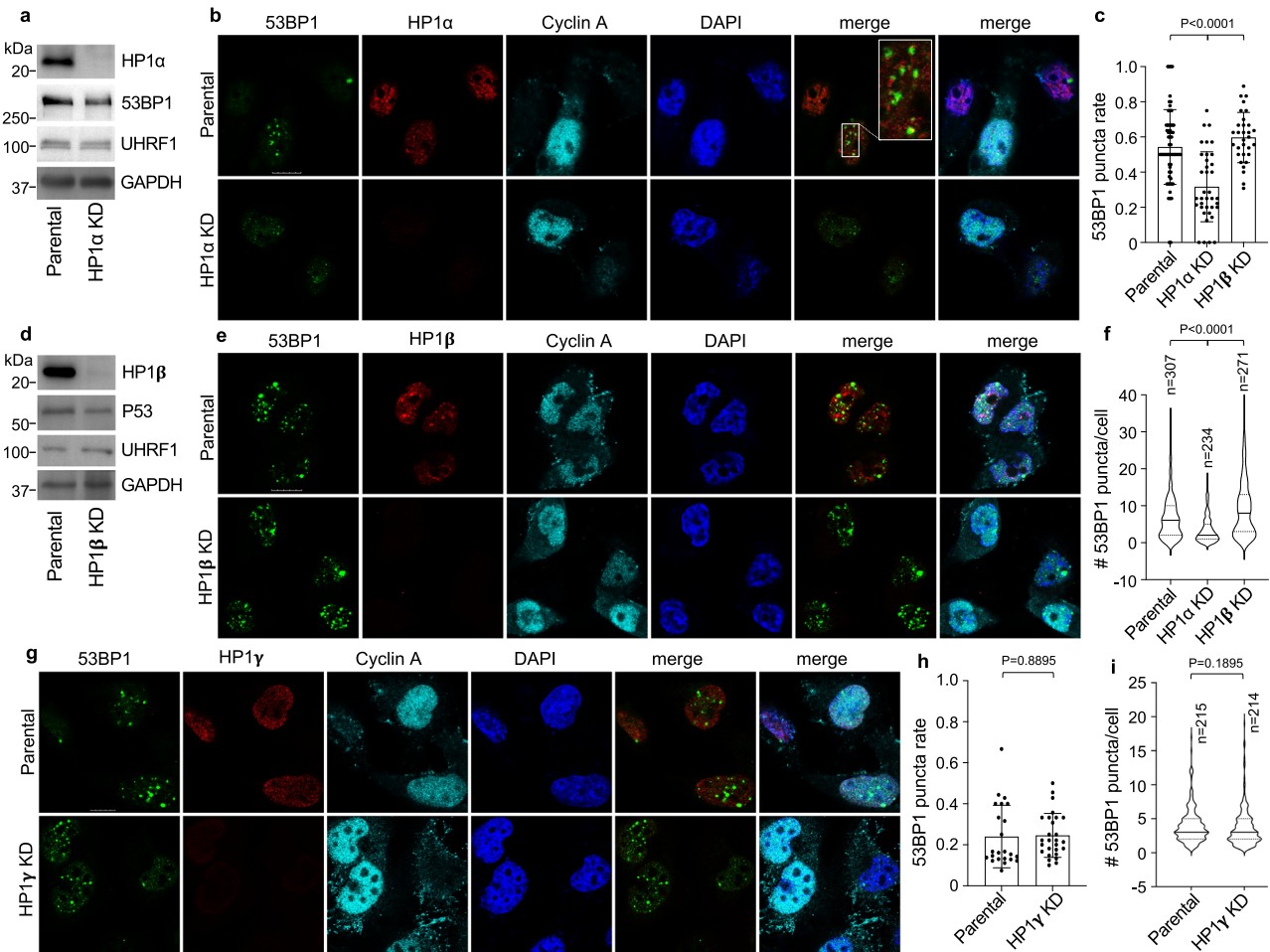

**Fig. 3 53BP1 puncta depend on heterochromatin.** MDA-MB-231 cells were stably depleted of HP1α (**a**) or HP1β (**d**), and protein expression was analyzed. Representative single z-plane confocal images of parental and HP1α (**b**), HP1β (**e**) or HP1γ (**g**) KD MDA-MB-231 cells. Quantitation of 53BP1 puncta rate (**c**, **h**) or number (**f**, **i**) from three independent experiments was shown. Violin plots of 53BP1 puncta number (**f**, **i**) represent mean, 25th, and 75th percentile from indicated number of cells. The whiskers extend to the minimum and maximum values. In **c**, $n = 60$, 37, and 31 images were analyzed for $n = 309$, 252 and 274 parental, HP1α KD and HP1β KD cells, respectively. In **h**, $n = 23$ and 27 images were analyzed from a total of $n = 215$ and 214 parental and HP1γ KD cells, respectively. Scale bar in all images is 10 μm. Data represent mean values and SD in (**c**) and (**h**). Unpaired two-tailed $t$ test using Prism 9.0 was performed for **c**, **f**, **h**, and **i** with 95% confidence intervals.

puncta (Fig. 3b, c, f), depletion of HP1β or HP1γ had almost no effect (Fig. 3c–i), suggesting that 53BP1 puncta formation mainly depends on HP1α.

To further confirm 53BP1's localization at heterochromatin, we performed ChIP-qPCR to determine the binding of 53BP1 to heterochromatic loci such as the AT-rich alpha satellite variants SATα, mcBox, and SATIII, an array of which could extend to mega-bases in length and is located at centromeric regions[42–44]. The results show that endogenous 53BP1 proteins were enriched at these loci in HEK293T cells, which were abolished by 53BP1 KD (Fig. 4a, b), confirming the antibody specificity. Importantly, re-expression of GFP-tagged 53BP1 full-length (FL)/wild type (WT) fully rescued heterochromatic DNA binding of 53BP1 in 53BP1 KD HEK293T cells (Fig. 4a, b), confirming that the binding is 53BP1 specific. Using the same approach, we found that the association of 53BP1 with SATIII was greatly reduced in HP1α-depleted cells (Supplementary Fig. 10h), supporting the role of HP1α in promoting 53BP1's association with heterochromatin. In all, these data suggest a constitutive association of 53BP1 with heterochromatin in the absence of DNA damage, which is consistent with its puncta formation.

**53BP1 is required for maintaining heterochromatin.** We showed that loss of 53BP1 resulted in a reduction in large heterochromatin centers in both human and mouse cells (Fig. 2), suggesting an important role of 53BP1 in maintaining the heterochromatin structure. Heterochromatin centers are highly condensed tandem repetitive DNAs characterized by transcriptional repression[42]. Hence, we assessed the transcription of heterochromatin using both 53BP1 KO and KD systems. Murine heterochromatin is categorized into major (Maj SAT) and minor (Min SAT) satellites at pericentric and centric chromosomes, respectively[45]. We found that *53bp1* KO MEFs expressed significantly higher levels of Maj and Min SATs than parental cells (Fig. 4c). Similarly, 53BP1 KD significantly increased levels of heterochromatic alpha satellite repetitive RNAs in human U-2 OS, HepG2, and HEK293T cells (Fig. 4d–f), indicating a general de-repression of heterochromatic transcription by 53BP1 KO or KD. Importantly, re-expressing 53BP1-FL/WT significantly reduced the elevated levels of heterochromatic satellite RNAs in 53BP1 KD HEK293T cells (Fig. 4e, f). These results support a role of 53BP1 in inhibiting aberrant transcription at heterochromatin. Last but not least, overexpression of human 53BP1-FL/WT restored the number of large heterochromatin centers in

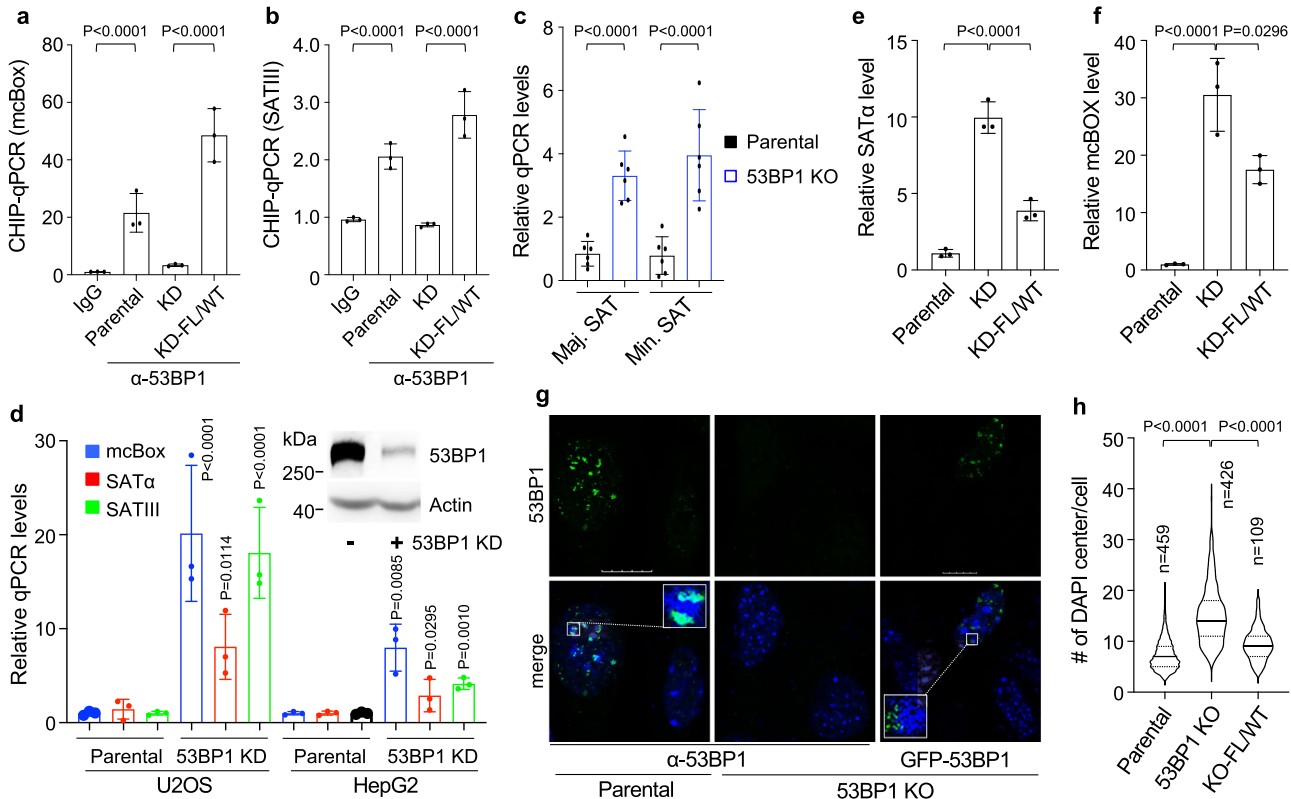

**Fig. 4 53BP1 maintains the repressive state of heterochromatin.** ChIP-qPCR of 53BP1 at alpha satellite repeats of mcBox (**a**) and SATIII (**b**) regions in HEK293T parental, 53BP1 KD and KD reconstituted with GFP-53BP1-FL/WT cells from $n = 3$ biological replicates. **c** Transcript levels of Maj and Min SATs measured by qPCR in parental and *53bp1* KO MEFs from $n = 6$ biological replicates collected from two independent experiments. **d** Transcript levels of alpha satellite repeats of heterochromatin from parental and stable 53BP1 KD U-2 OS or HepG2 cells from $n = 3$ biological replicates. Inset: representative protein expression in U-2 OS cells from two independent experiments. Transcript levels of the alpha satellite repeat SATα (**e**) and mcBOX (**f**) from HEK293T parental, 53BP1 KD and KD cells reconstituted with GFP-53BP1-FL/WT from $n = 3$ biological replicates. **g** Representative single z-plane confocal images of parental, *53bp1* KO and KO MEFs reconstituted with GFL-53BP1-FL/WT. Squares: enlarged areas. Scale bar is 10 μm. **h** Violin plot of the number of heterochromatin centers visualized by DAPI from indicated number of cells from at least three independent experiments. Data represent mean, 25th, and 75th percentiles with the whiskers extending to the minimum and maximum values. Data represent mean values and SD in (**a–f**). Statistical analyses in (**d**) were compared with parental cells. Unpaired two-tailed *t* test was done by Prism 9.0 with 95% confidence intervals in (**a–f**) and (**h**).

*53bp1* KO MEFs (Fig. 4g, h). Together, these data support a previously uncharacterized role of 53BP1 in maintaining both the structural integrity and the transcriptional repression of heterochromatin.

**53BP1 domains required for puncta formation.** Next, we decided to map domains of 53BP1 that are responsible for the puncta formation. Various GFP-53BP1 constructs (Supplementary Fig. 11a) were generated and their expression was confirmed in U-2 OS cells (Supplementary Fig. 11b). For cells expressing similar levels of GFP proteins (Supplementary Fig. 11c), GFP-53BP1-FL/WT formed puncta in about 10% of U-2 OS cells in this experiment (Supplementary Fig. 11d). The fragments lacking an NLS (N2, C2, C3) were expressed in the cytoplasm (Supplementary Fig. 11c) and therefore were excluded from further study. Deletion of the C-terminal BRCTs and/or the N-terminus (both are involved in DSB repair) did not affect 53BP1 puncta formation (Supplementary Fig. 11c, C and C1 fragments), indicating that puncta formation is a distinct activity of 53BP1 from its role in DSB repair. These data also suggest that the OD-Tudor-UDR-NLS region (i.e., the C1 fragment) is responsible for puncta formation of 53BP1, which overlaps with the DSB foci-forming domain; however, results later will confirm the independence of these two functions.

We noted that 53BP1-C1 formed much higher rate of puncta than 53BP1-FL (Supplementary Fig. 11d), and U-2 OS cells expressing 53BP1-C1 puncta also had increased number of DAPI puncta (Supplementary Fig. 11e); further, these DAPI puncta resembled the large heterochromatin centers seen in MEFs (Supplementary Fig. 11e), which have not been observed in normal human cells. The median, 25%, and 75% percentiles of the GFP-53BP1-C1 puncta size were 5.081, 3.290, and 7.330 μm$^2$ (in area), respectively (Supplementary Fig. 11f), which were significantly larger than those formed by endogenous 53BP1. Also, the number of 53BP1-C1 puncta was much higher than that of 53BP1-FL (Supplementary Fig. 11g). To determine if the puncta formed by 53BP1-C1 also localize at heterochromatin, we examined the co-localization of GFP-53BP1-C1 with heterochromatin markers. We found that the percentage of 53BP1-C1 puncta that co-localized with heterochromatin markers was significantly higher than that of 53BP1-FL (Supplementary Fig. 12a). These results further confirm that the C1 fragment is responsible for 53BP1 puncta formation at heterochromatin. Yet, like endogenous 53BP1, 53BP1-C1 puncta did not form at transcriptional sites while showing limited co-localization with telomere ends (Supplementary Fig. 12a). Cells with 53BP1-C1 puncta did not show an increase in foci formation for pKAP1 under normal growth conditions (Supplementary Fig. 12a), indicating that 53BP1-C1 puncta, like endogenous

ones, did not induce or represent DNA damage. Further, 53BP1-C1 puncta-positive U-2 OS cells did not express any elevated levels of cleaved Caspase 3 (Supplementary Fig. 12a), suggesting that the highly condensed DNA in these cells was not due to apoptotic cell death. Together, these results suggest that while 53BP1-C1 is a gain-of-function mutant of puncta formation, the puncta formed by 53BP1-C1 retain all the properties of those formed by 53BP1-FL at heterochromatin. Hence, the 53BP1-C1 mutant serves as a great tool to dissect the difference between DSB foci and puncta of 53BP1.

The C1 region has been reported to be responsible for forming 53BP1 foci at DSB sites[1,2]. We confirmed this by detecting a strong co-localization between GFP-53BP1-C1 'dots' and γH2AX foci in ~40% transfected cells after treatment with DNA damage (Supplementary Fig. 12b, lower). However, we also observed that 53BP1-C1 'dots' in other ~40–50% of cells had no or minimal co-localization with γH2AX foci (Supplementary Fig. 12b, upper). The 53BP1-C1 'dots' that had limited co-localization with γH2AX in the presence of DNA damage displayed the same pattern (shape, size and number) as 53BP1-C1 puncta formed in the absence of DNA damage. Using co-localization with γH2AX foci as a readout, we were able to find that DNA damage did not affect puncta formation of 53BP1-C1 (Supplementary Fig. 12c), supporting the idea that 53BP1 puncta are independent of DSB repair. This feature also allowed us to discriminate between the puncta and DSB foci of 53BP1 by examining their co-localization with γH2AX or other DSB markers: when there is substantial co-localization, we consider the 53BP1 'dots' as DSB foci; otherwise, they will be counted as 53BP1 puncta.

**53BP1 puncta undergo LLPS with HP1α**. Next, we wanted to further understand the biological significance of the 53BP1 puncta. Endogenous 53BP1 didn't form puncta in mitosis (Fig. 1b, e). Live cell imaging revealed that 53BP1-C1 puncta persisted throughout interphase, quickly resolved during mitosis, but rapidly re-formed when cells entered G1 phase (Supplementary Fig. 13a), like those of *Drosophila* HP1α's liquid droplets[32,33]. Since 53BP1 puncta co-localized with HP1α at heterochromatin, we hypothesized that these puncta may also represent LLPS. Proteins that can phase separate often contain disordered peptide sequences. Using the IUPRED2A prediction tool[46], we identified intrinsically disordered regions throughout the 53BP1 peptide (Supplementary Fig. 13b). Liquid droplets are sensitive to 1,6-hexanediol, an aliphatic alcohol that disrupts weak inter-molecular hydrophobic interactions required for LLPS[47]. Treatment of cells with 1,6-hexanediol significantly reduced puncta formation of endogenous 53BP1 (Supplementary Fig. 13c, d). Further, 1,6-hexanediol treatment reduced the puncta formed by GFP-53BP1-C1, mCherry-HP1α or both in live cells (Supplementary Fig. 13e, f). We noticed that 1,6-hexanediol did not completely abolish 53BP1 puncta, which is similar to its effect on the droplets formed by HP1α[48]. Nonetheless, the reduction in 53BP1 puncta (both the rate and the number) was significant, supporting the idea that these 53BP1 puncta represent liquid droplets that were phase separating at heterochromatin.

To determine if 53BP1 indeed undergoes LLPS, we first purified mGFP-53BP1-C1 and mCherry-HP1α proteins from bacteria (Fig. 5a and Supplementary Fig. 14a) and performed in vitro LLPS assay without the addition of any crowding agents that may cause artificial liquid droplet formation. As previously reported[32], un-phosphorylated HP1α alone did not form liquid droplets (Supplementary Fig. 14b). 53BP1-C1 alone at 10 μM only formed a few very tiny droplets (Supplementary Fig. 14b). However, combining HP1α and 53BP1-C1 readily promoted protein precipitation after centrifugation (Fig. 5b, red circles),

indicating the formation of particles by these two proteins in solution. When examining the mixture under fluorescence microscopy, we found that they formed both large hollow sphere-like and small solid circle-like liquid droplets (Fig. 5c). Further, 53BP1-C1 and HP1α formed liquid droplets in vitro in a concentration-dependent manner (Supplementary Fig. 14c, d). These results support the conclusion that 53BP1-C1 and HP1α can phase separate in vitro in a mutually dependent manner.

To narrow down the region required for 53BP1's liquid droplet formation with HP1α, we compared the effects of 53BP1-C1 with two additional constructs (53BP1-C2 and -C3) (Fig. 5a and Supplementary Fig. 14a) on liquid droplet formation with HP1α in vitro. The results show that 53BP1-C2 or -C3, but not GFP control, induced HP1α precipitation after centrifugation (Fig. 5b, red circles). Further, HP1α and 53BP1-C2, and to a lesser degree C3, formed liquid droplets in vitro (Fig. 5c). These data suggest that the OD is the minimal region required for promoting liquid droplet formation of 53BP1 with HP1α.

To test this idea in vivo, we assessed the mobility of GFP-HP1α or GFP-53BP1-C1 puncta in cells by fluorescence recovery after photobleaching (FRAP), an approach commonly used to evaluate LLPS in vivo. Like endogenous proteins, GFP-HP1α and GFP-53BP1-C1 formed much less puncta in 53BP1 KO MCF-10A (Fig. 5d) and HP1α KD U-2 OS cells (Fig. 5f), respectively. Nonetheless, the remaining GFP-HP1α and GFP-53BP1-C1 puncta in these cells allowed us to assess their mobility. The results show that both GFP-HP1α and GFP-53BP1-C1 puncta quickly recovered after photobleach with similar kinetics in parental MCF-10A (Fig. 5d, e) or U-2 OS cells (Fig. 5f, g), which is consistent with that of *Drosophila* HP1α[33]. Remarkably, 53BP1 KO or HP1α KD significantly reduced the FRAP of GFP-HP1α or 53BP1-C1 puncta (Fig. 5d–g). Of note, we observed unstable recovery of 53BP1-C1 puncta in cells depleted of HP1α (Fig. 5f, the arrow), further supporting the role of HP1α in stable recruitment of 53BP1 to heterochromatin.

Further, we examined FRAP of co-localized puncta for mCherry-HP1α and GFP-53BP1-C1 in U-2 OS cells. The results show that HP1α recovered faster than 53BP1-C1 (Supplementary Fig. 15a, b), like the observation between HP1α and SUV39H1[35], indicating a leading role of HP1α in regulating LLPS of other heterochromatin proteins. Together, these in vitro and in vivo data strongly suggest that 53BP1 puncta at heterochromatin represent liquid droplets that undergo phase separation, which depends on, but also stabilize HP1α.

**LLPS of 53BP1 is independent of DSB repair**. Our data so far suggest that 53BP1 puncta formed at heterochromatin are different from DSB foci. Yet, recent studies suggested that 53BP1 could undergo LLPS at DSB site[49,50]. Hence, we decided to further investigate if 53BP1 LLPS at heterochromatin is related to DSB repair. To answer this question, we compared detailed roles of 53BP1 domains and residues involved in LLPS and DSB foci by generating additional 53BP1 mutants (adding an NLS when needed) (Fig. 6a). Like C1, C2-NLS formed puncta (Fig. 6b, c), indicating that the UDR is not required for LLPS although it is important for DSB foci[7]. When the OD was deleted (Fig. 6a, C4), the puncta formation was not detected (Fig. 6b, c), suggesting that OD is absolutely essential, which is consistent with the in vitro LLPS results (Fig. 5c). However, the OD only (Fig. 6a, C3-NLS) formed much less puncta than C2-NLS (Fig. 6b, c), suggesting that while the Tudor domain is not essential, it is required for the maximal level of 53BP1 puncta formation in vivo. In contrast, none of the C2-NLS, C3-NLS, or C4 formed γH2AX co-localizing DSB foci after DNA damage (Fig. 6d), confirming the loss of their DSB repair function.

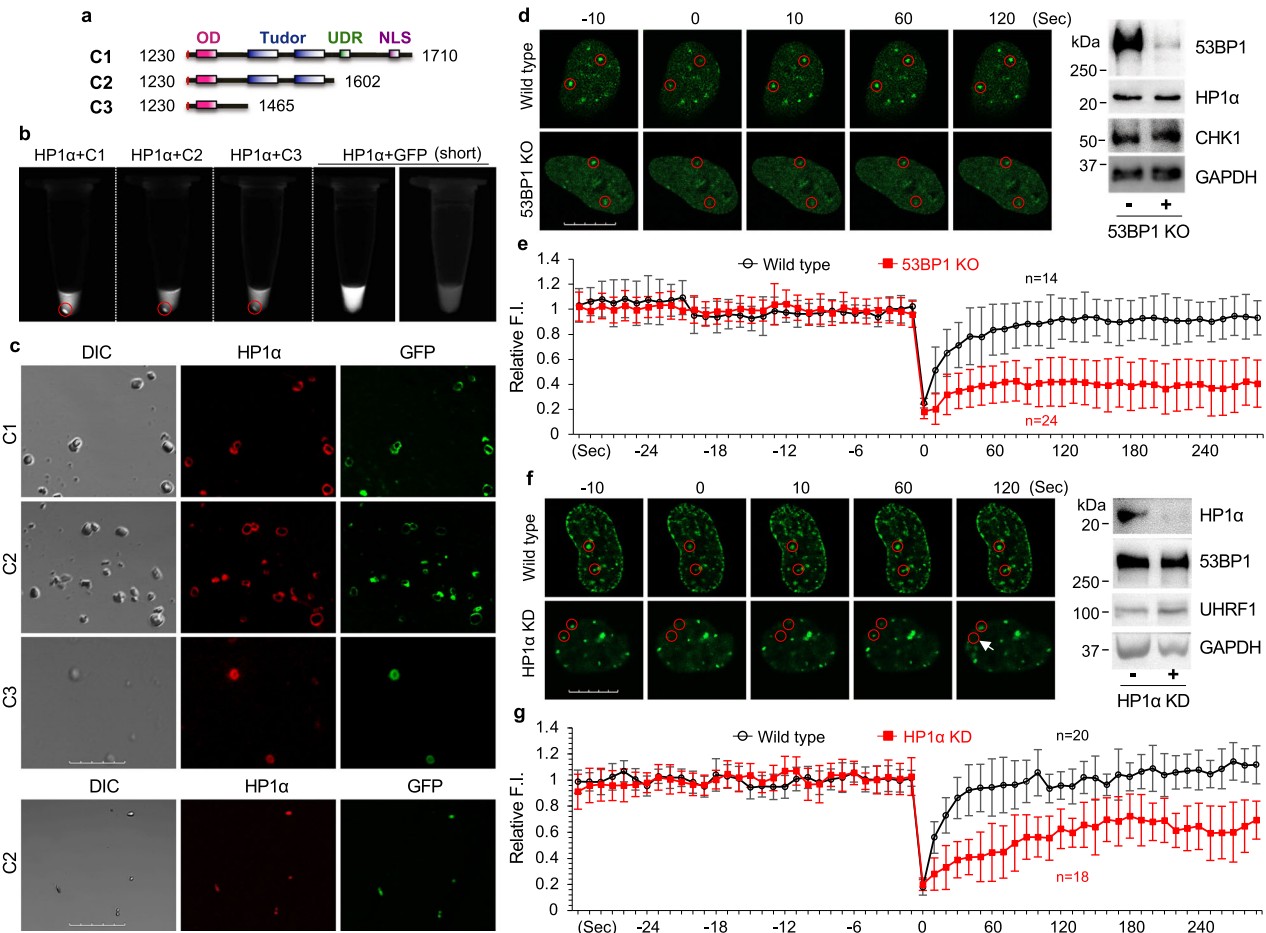

**Fig. 5 LLPS of 53BP1 and HP1α. a** Generation of GFP-53BP1 fragments. **b** Purified proteins were mixed on ice, incubated at room temperature for 20 min, centrifuged at 10,000 g for 10 min and visualized by the Tanon 2500 Imager at 590 nm. Far Right: short exposure for the mixture of HP1α and GFP. Red circles indicate precipitations after centrifugation. **c** Representative bright field and single z-plane confocal images of liquid droplets formed by mGFP-53BP1-C1, -C2 or -C3 with mCherry-HP1α in vitro. *Lower panel*: smaller droplets. Scale bar is 40 μm. **d** MCF-10A parental or 53BP1 KO cells were transfected with GFP-HP1α for 48 h and FRAP was performed. Representative single z-plane confocal images are shown. *Right:* protein expression. The residual 53BP1 signal in the KO lane might have come from loading contamination. **e** FRAP analysis of cells in (**d**) from n = 14 and 24 events. **f** U-2 OS parental or HP1α KD cells were transfected with GFP-53BP1-C1 for 48 h and FRAP was performed. Representative single z-plane confocal images are shown. Right: protein expression. The arrow indicates a 53BP1-C1 punctum that transiently recovered (60 s) but then disappeared (120 s). **g** FRAP analysis of cells in (**f**) from n = 20 and 18 events. The X-axis (time) in (**e**) and (**f**) is not scaled and 0 indicates photobleaching. Red circles in (**d**) and (**f**) indicate puncta that were bleached. Scale bars in (**e**) and (**f**) are 10 μm. FRAP quantitation in (**e**) and (**g**) represent mean values and SD acquired from two independent experiments.

The Asp 1521 (D1521) residue in the Tudor domain is essential for 53BP1's DSB repair function through binding to H4K20me[3–6]. Consistently, the 53BP1-C1/D1521R mutant failed to form γH2AX co-localizing DSB foci (Fig. 6d); however, it still formed puncta as strongly as 53BP1-C1 (Fig. 6b, c). *S. pombe* Crb2 shares similar domains with human 53BP1-C except the lack of the UDR (Fig. 6a). Interestingly, codon-optimized GFP-Crb2 formed puncta in U-2 OS cells (Fig. 6e, f), and mutating D402, the *pombe* residue corresponding to human D1521, did not affect GFP-Crb2's puncta formation (Fig. 6e, f). These data confirm that binding to H4K20me or H2AK15ub (both required for DBS foci) is dispensable for puncta formation of 53BP1, supporting the model in which LLPS is uncoupled from the canonical DSB repair of 53BP1.

The OD of 53BP1 shares four conserved residues with Crb2 (Fig. 6a). While single Ala mutations had no impact, mutating multiple residues, while not affecting their expression levels (Fig. 6g and Supplementary Fig. 10i), gradually reduced puncta formation of 53BP1-C1, and all four residues mutated (53BP1-C1-OD/4A) showed the greatest reduction (Fig. 6g, i).

Consistently, 53BP1-FL/OD-4A interacted weaker with FLAG-HP1α than 53BP1-FL/WT or 53BP1-FL/DR (Supplementary Fig. 10j). Similarly, the 53BP1-C1/OD-4A mutant rarely interacted with FLAG-HP1α, like the mutant lacking the OD (C4, Supplementary Fig. 10k). These results highlight the importance of these four residues in 53BP1's puncta/LLPS. In contrast, the 53BP1-C1/OD-4A mutant formed γH2AX co-localizing DSB foci as strongly as 53BP1-C1 (Fig. 6h), indicating that these residues are dispensable for DSB repair, which will be further confirmed below.

Treatment with an ATM inhibitor did not affect 53BP1-C1's puncta formation (Fig. 7a), suggesting that ATM-dependent phosphorylation is not required for LLPS although it is essential for DSB repair[10]. These are consistent with the results obtained from 53BP1 mutants depleted of the N-terminus (Supplementary Fig. 11c, d). To further test this idea, we reconstituted 53BP1 KO U-2 OS cells with various GFP-53BP1 constructs that were expressed at comparable levels with the exception of 53BP1-C4 (Fig. 7b), and examined the formation of 53BP1 puncta or DSB foci. Under normal growth conditions, 53BP1-FL/WT and -FL/

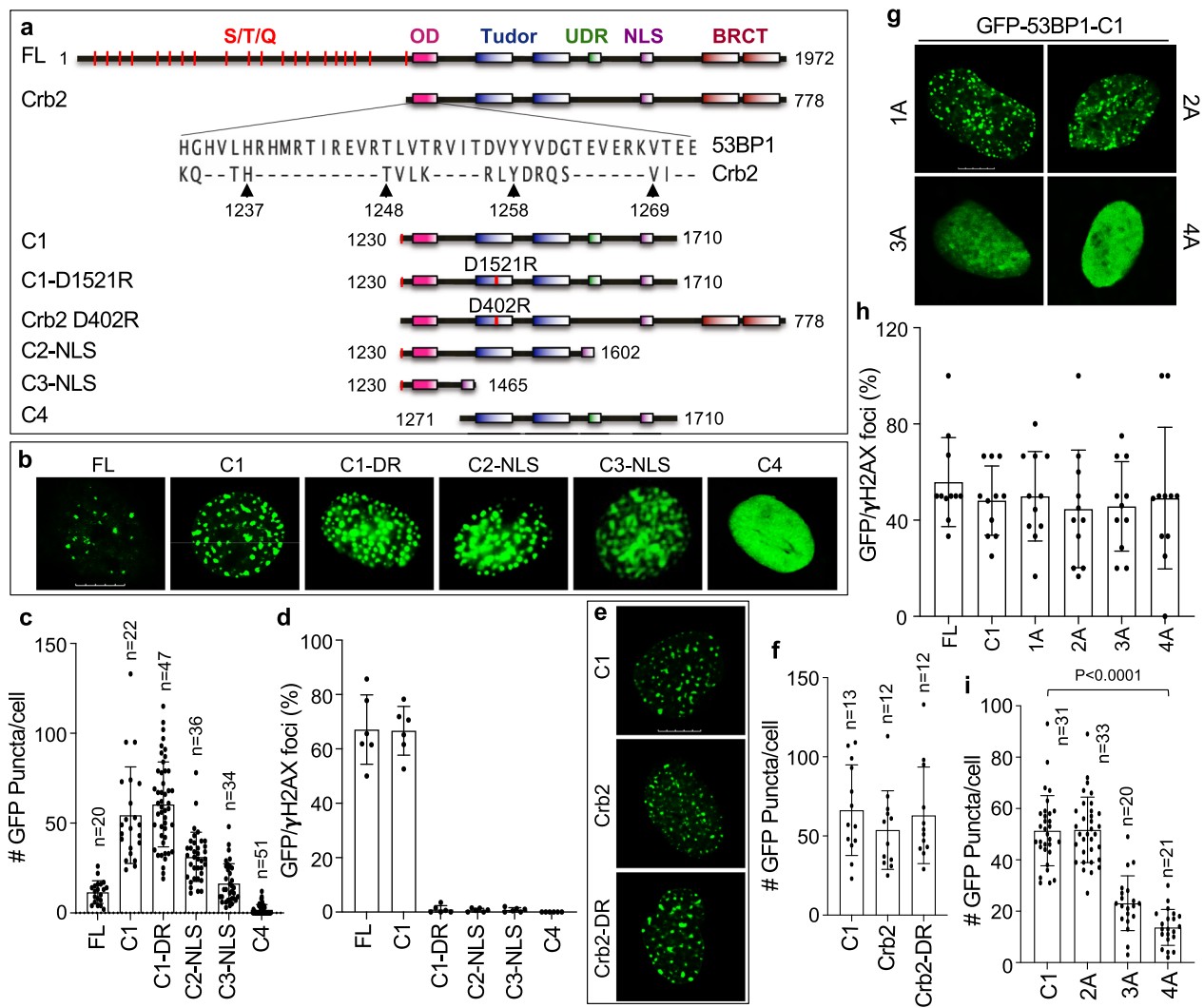

**Fig. 6 Domains and residues of 53BP1 involved in LLPS and DSB repair. a** Schematic diagram for mutant generation and alignment of the OD between 53BP1 and Crb2. Conserved residues (H1237, T1248, Y1258, and V1269 in human 53BP1) in the OD are indicated by arrowheads. **b** Representative single z-plane confocal images of 53BP1 puncta formed by different constructs in U-2 OS cells in the absence of DNA damage. **c** Puncta number of GFP-53BP1 constructs analyzed from indicated number of cells from two independent experiments. **d** U-2 OS cells were transfected with GFP-53BP1 constructs for 48 h, irradiated with 5 Gy of IR, fixed 4 h later and stained with anti-γH2AX antibodies. The percentage of cells showing co-localizing foci for GFP and γH2AX over total GFP positive cells was assessed from $n = 6$ biological replicates acquired from two independent experiments. **e** Representative single z-plane confocal images of puncta formed by GFP-53BP1-C1, GFP-Crb2, and GFP-Crb2/D402R in U-2 OS cells in the absence of DNA damage. **f** Puncta number of GFP-53BP1-C1, GFP-Crb2, and GFP-Crb2/D402R analyzed from indicated number of cells from two independent experiments. **g** Representative single z-plane confocal images for puncta formed by GFP-53BP1-C1/OD mutants in U-2 OS cells in the absence of DNA damage. 1A-4A indicates Ala substitution at the four conserved OD residues. **h** U-2 OS cells were transfected with GFP-53BP1 constructs for 48 h, treated with 3.5 μM bleomycin for 6 h, fixed, and stained with anti-γH2AX antibodies. The percentage of cells showing co-localizing foci for GFP and γH2AX over total GFP positive cells was assessed from $n = 11$ images from two independent experiments. **i** Puncta number of cells in (**g**) in the absence of DNA damage was analyzed from indicated number of cells. Data represent mean values and SD in (**c**), (**d**), (**f**), (**h**) and (**i**). Scale bar is 10 μm. Unpaired two-tailed t test was performed in Prism 9.0 with 95% confidence intervals in (**i**).

DR completely rescued 53BP1's puncta in 53BP1 KO cells, which was greatly enhanced by 53BP1-C1 (Fig. 7c). In contrast, 53BP1-FL/OD-4A or 53BP1-C4 failed to form puncta (Fig. 7c).

Then we determined the DSB repair capability of these constructs by assessing bleomycin-induced foci formation of RIF1, a key downstream factor of 53BP1 in DSB repair[1,2]. To discriminate the puncta from DSB foci of 53BP1, we asked whether the 'dots' formed by 53BP1 co-localize with RIF1 foci in the presence of DNA damage. If they co-localize, we consider the 53BP1 dot as a DSB focus; otherwise, it will be counted as a 53BP1 punctum, as we previously described for γH2AX (Supplementary Fig. 12b). Using GFP-53BP1-C1 as the model, we measured the size of 53BP1 puncta and DSB foci side by side and found that 53BP1-C1 puncta were significantly larger than the DSB foci (the median size of 53BP1-C1 puncta and DSB foci was 3.120 and 0.377 μm², respectively, Supplementary Fig. 14e). Using these criteria, we found that in parental cells, RIF1 and 53BP1 formed foci at the same nuclear location after bleomycin treatment (Fig. 7d), which was abolished by 53BP1 KO (Fig. 7d, e), indicating the loss of 53BP1's DSB repair function in KO cells. GFP-53BP1-FL/WT and GFP-53BP1-FL/OD-4A rescued RIF1's foci (Fig. 7d, e), which not only confirmed the restoration of the 53BP1-dependent DSB repair signaling in these cells, but also further supported our idea that the OD-4A mutation did not impair 53BP1's DSB repair function. In

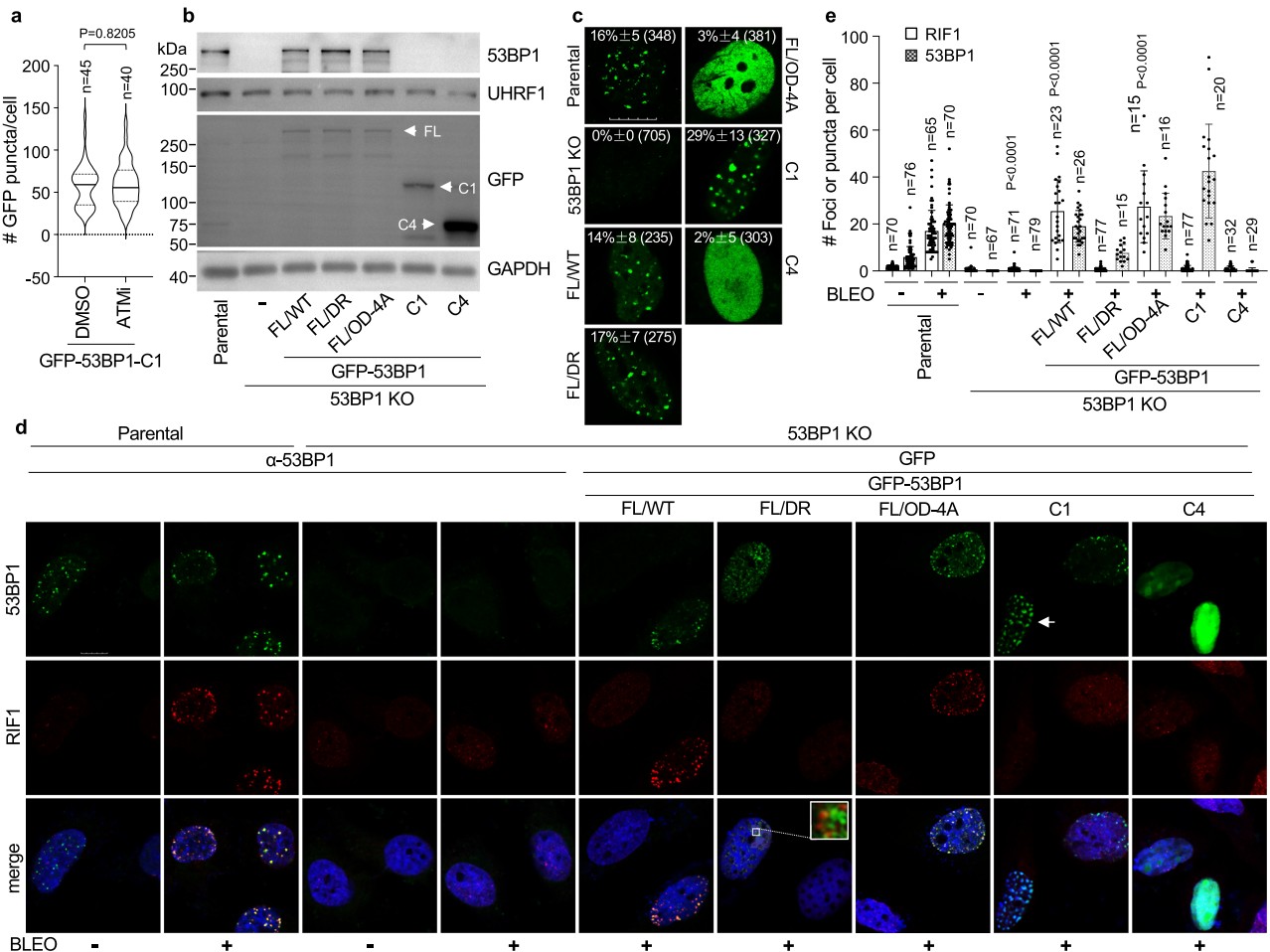

**Fig. 7 Uncoupling between LLPS and DSB repair of 53BP1. a** U-2 OS cells were transfected with GFP-53BP1-C1 for 36 h, treated with 500 nM ATM inhibitor (ATMi) for 12 h, and the number of GFP-53BP1-C1 puncta per cell was counted and presented as Violin plot from indicated number of cells from two independent experiments. Data represent mean, 25th, and 75th percentiles with the whiskers extending to the minimum and maximum values. **b** 53BP1 KO U-2 OS cells re-constituted with GFP-53BP1 constructs and protein expression was examined. The uncropped blots are provided in the Source Data file. **c** Representative single z-plane confocal images of 53BP1 puncta in the absence of DNA damage. Quantitation of the percentage of GFP puncta positive cells from n = 18, 28, 33, 31, 30, 37, and 34 images for each group were analyzed from indicated number (in parenthesis) of cells and presented as mean values and SD acquired from at least two independent experiments. **d** U-2 OS cells from (**b**) were treated with 3.5 μM bleomycin for 6 h, fixed and stained with the anti-RIF1 antibody. Representative single z-plane confocal images are shown. Green signals in non-transfected cells represent endogenous 53BP1 by anti-53BP1 antibody staining. Square: an enlarged area showing rare co-localization between 53BP1-FL/DR and RIF1 foci, which helps determine this cell being 53BP1-FL/DR puncta positive, but DSB foci negative. Arrow: a cell showing 53BP1-C1 forming liquid droplets, but not DSB foci determined by the puncta size. Scale bar in all images is 10 μm. **e** Foci or puncta number of RIF1 and 53BP1 from indicated number (n) of cells in (**d**). Data represent mean values and SD from two independent experiments. Unpaired two-tailed t test was performed in Prism 9.0 with 95% confidence intervals in (**a**) and (**e**). In (**e**), while the 53BP1 KO group was compared to the parental group in the presence of BLEO, others were compared to the 53BP1 KO group in the presence of BLEO for statistical analysis.

contrast, GFP-53BP1-FL/DR, GFP-53BP1-C1 or GFP-53BP1-C4 failed to rescue RIF1's foci formation (Fig. 7d, e), confirming that they are defective in DSB repair, similar to the findings for 53BP1-C1 in a previous report[51].

To further determine the LLPS function of these 53BP1 constructs, we performed FRAP to analyze the mobility of mCherry-HP1α in these reconstituted cell lines. The results show that while 53BP1-FL/WT, -FL/DR, and -C1 rescued the mobility of mCherry-HP1α in 53BP1 KO U-2 OS cells, 53BP1-FL/OD-4A or -C4 did not (Supplementary Fig. 15c, d). Together, these data identify separation-of-function mutants of 53BP1 in DSB repair (FL/OD-4A) and LLPS (FL/DR or C1), respectively, supporting the independence of these two functions.

**53BP1's LLPS function protects cells from stress-induced DNA damage and senescence.** Loss of heterochromatin may result in

genomic instability and cellular senescence[24–26]. Given the critical role of 53BP1 in heterochromatin maintenance, we asked if the LLPS function of 53BP1 is involved in the protective effect of heterochromatin against genomic instability and senescence. We first performed an alkaline comet assay to determine bleomycin-induced DNA damage in the above-generated U-2 OS cell lines. Bleomycin treatment increased DNA damage in parental cells, which was further significantly increased by 53BP1 KO, indicating enhanced genomic instability when 53BP1 is depleted (Fig. 8a, b). However, reconstitution of 53BP1-FL/WT significantly inhibited the elevated DNA damage in 53BP1 KO cells (Fig. 8a, b). A less strong but significant rescue was also observed for 53BP1-FL/DR and 53BP1-C1 (Fig. 8a, b). 53BP1-FL/OD-4A also significantly reduced DNA damage in 53BP1 KO cells (Fig. 8a, b), likely because this mutant retained the DSB repair function. In contrast, 53BP1-C4 failed to reduce the increased DNA damage in 53BP1 KO cells

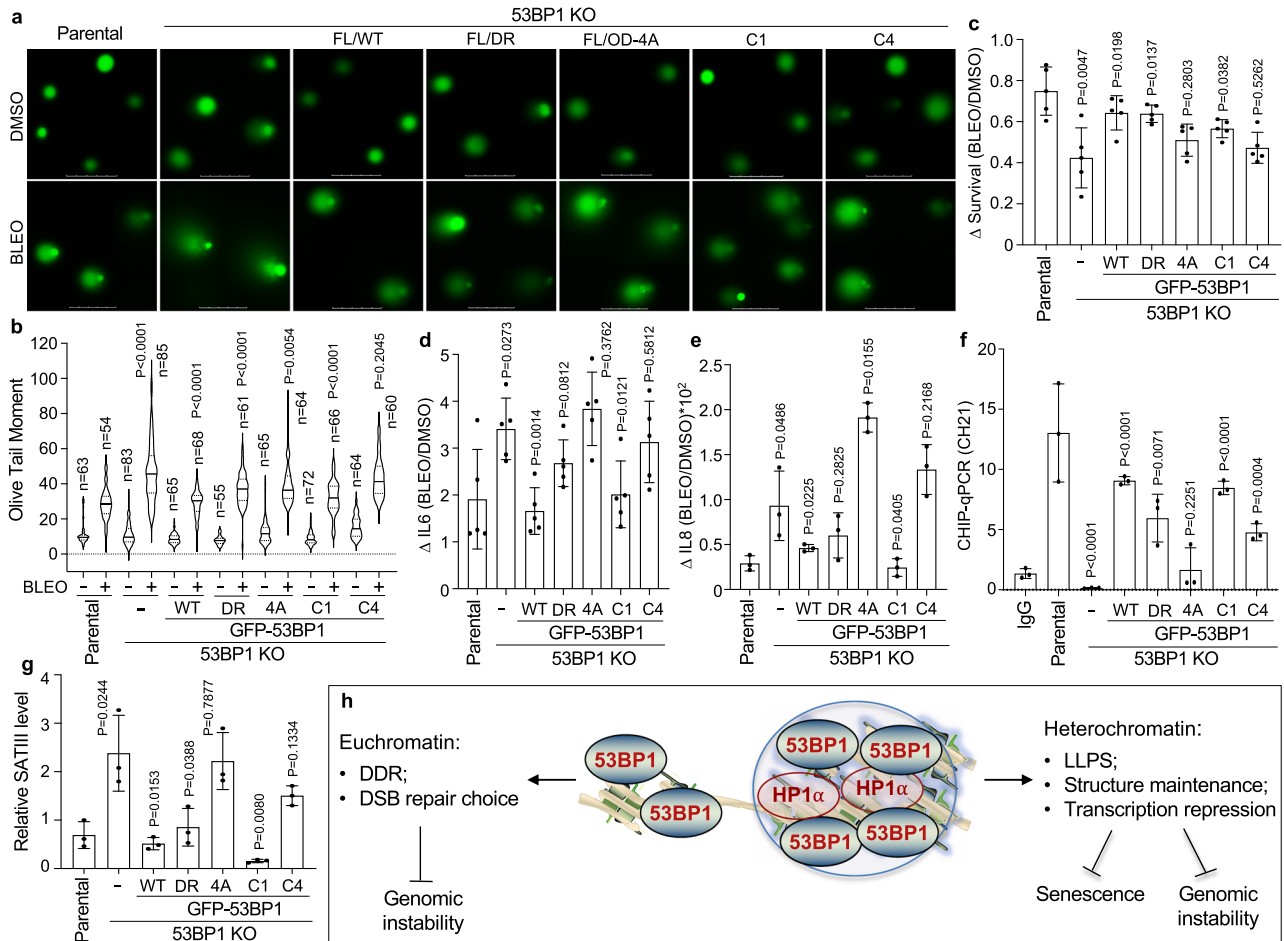

**Fig. 8 The protective role of 53BP1's LLPS function. a** Representative projection images of the comet assay from U-2 OS parental, 53BP1 KO and KO cells stably reconstituted with various 53BP1 constructs after a treatment with 3.5 μM bleomycin (BLEO) for 6 h. Scale bar is 100 μm. **b** Violin plot of the Olive Tail Moment from indicated number (*n*) of cells in (**a**). Data represent mean, 25th, and 75th percentile of the BLEO/DMSO ratio with the whiskers extending to the minimum and maximum values obtained from two independent experiments. **c** Survival difference of U-2 OS cell lines after treatment with 3.5 μM BLEO for 6 h. Data represent mean values and SD from *n* = 5 biological replicates. ELISA of IL-6 (**d**) and IL-8 (**e**) levels from U-2 OS cell lines after treatment with 0.35 μM BLEO for 7 days. Data represent mean values and SD of the level difference (BLEO/DMSO ratio) for each group normalized to cell numbers from *n* = 5 (**d**) and *n* = 3 (**e**) biological replicates. ChIP-qPCR (**f**) and qPCR (**g**) from U-2 OS cells. Data represent mean values and SD normalized to protein levels of 53BP1 or GFP-53BP1 from *n* = 3 biological replicates. Unpaired two-tailed *t* test was performed by Prism 9.0 with 95% confidence intervals in (**c–g**). Except for the 53BP1 KO group that was compared with the parental group, other groups were compared to the KO group individually for statistical analysis. **h** Proposed model. See Discussion for more detail.

(Fig. 8a, b), confirming its loss of both DSB repair and LLPS function.

To further determine the protective role of 53BP1's LLPS function against genomic instability, we measured the long-term survival capability of the above-generated cell lines after DNA damage. Deletion of 53BP1 greatly sensitized cells to bleomycin (Fig. 8c), which was significantly rescued by GFP-53BP1-FL, and less strongly but significantly by GFP-53BP1-FL/DR and GFP-53BP1-C1, but not by GFP-53BP1-FL/OD-4A or GFP-53BP1-C4 (Fig. 8c). When we examined the secretion of senescence-associated inflammatory cytokines such as IL-6 and IL-8 after chronic exposure to bleomycin, we found that 53BP1 KO greatly increased the levels of these inflammatory factors, which were significantly suppressed by 53BP1-FL/WT or -C1 (Fig. 8d, e). FL/DR moderately reduced the increased levels of IL6 and IL8 (Fig. 8d, e); however, FL/OD-4A or -C4 showed no protective effect (Fig. 8d, e). Further, 53BP1-FL/WT, -FL/DR or -C1, but not FL/OD-4A, rescued the binding of 53BP1 on a heterochromatic alpha satellite array on chromosome 21 (CH21) (Fig. 8f) and significantly reduced the increased transcription of

heterochromatic repetitive DNA in 53BP1 KO cells (Fig. 8g). The 53BP1-C4 mutant was found to localize at heterochromatin probably due to its high expression (Fig. 8f); yet, it was largely defective in suppressing heterochromatin transcription (Fig. 8g). Together, these results suggest that 53BP1's LLPS function protected cells from DNA damage and genomic instability, at least partially, through maintaining/stabilizing heterochromatin (Fig. 8h).

## Discussion

53BP1 has been primarily known as an important DSB repair choice mediator[2]. However, not all of its functions seem to be fully explained by the DSB repair activity. For instance, *53bp1*$^{-/-}$ MEFs growing under normal conditions tended to be aneuploid and tetraploid[36], displaying chromosome segregation errors during mitosis similar to cells lacking HP1α[52]. Here we provide several lines of evidence to show an unexpected, yet important, role of 53BP1 in maintaining heterochromatin structure and function, and consequently genome stability through LLPS

(Fig. 8h). First, 53BP1 formed puncta at heterochromatin in a way dependent on the key heterochromatin protein HP1α and these puncta bear properties of LLPS. Second, 53BP1, in turn, stabilized the retention and mobility of HP1α at heterochromatin. Third, loss of 53BP1 led to reduced structural integrity and de-repression of heterochromatin, accompanied with increased sensitivity to DNA damage. Most importantly, reconstitution of 53BP1 KO cells with LLPS proficient (regardless of the DSB repair proficiency), but not LLPS deficient 53BP1 restored the heterochromatin structure and transcriptional repression, as well as reduced DNA damage and senescence caused by DNA damage in 53BP1 KO cells.

At this moment, a couple of possibilities could be considered for the detailed molecular mechanisms underlying the protective effect of 53BP1's LLPS function. First, by stabilizing the membraneless compartment at heterochromatin, 53BP1's LLPS might help preserve the genome-wide chromosomal stability. This is supported by our data showing that 53BP1 puncta often abutted or wrapped around heterochromatin centers, and depletion of 53BP1 led to de-repression of heterochromatin, which was rescued by 53BP1 constructs proficient for LLPS function. Another possibility is that 53BP1's LLPS activity might facilitate the movement of DSB ends generated at heterochromatin from the center to the periphery of the nucleus, where they will be repaired[53–58]. 53BP1 has been reported to direct the movement/mobility of DSB ends induced by IR or telomere unprotection[51,59], although it is not clear whether this also involves a LLPS function. Even though the LLPS activity of 53BP1 is independent of its DSB repair function, it does not preclude the possibility that LLPS could help direct the DSB end movement at heterochromatin to facilitate the repair. In addition, our data show that 53BP1 puncta were formed in the actively dividing stage of the cell cycle (i.e., Cyclin A positive cells). It was reported that 53BP1 is expelled from the damage site by BRCA1 when the CDK kinase activity is high[60,61]. Hence, our findings indicate that puncta formation of 53BP1 might inhibit NHEJ while promoting HR-dependent DSB repair in S to G2 phases, a surprising idea that is opposite to the currently recognized function of 53BP1 in DSB repair choice yet intriguing enough that warrants further investigation.

53BP1 has been previously reported to form nuclear bodies in G1 phase under normal growth conditions, which represented spontaneous DNA damage resulting from errors in the previous round of the cell division cycle[37]. However, whether all these dots/puncta represented DNA damage has not been investigated, and if not, what is the nature of these puncta remained unknown. In the current study, we show that not all 53BP1 puncta represent DNA damage under normal growth condition; rather, a significant portion of them were localized at heterochromatin, behaved like liquid droplets, and depended on heterochromatin factors to form. At first glance, 53BP1 puncta only partially (~40%) co-localized with heterochromatin markers including HP1α and H3K9me3, which might cast doubt on its significance. However, our extensive analyses showed that such partial co-localization is significant in maintaining both the structural integrity and the transcriptional repression of heterochromatin. Hence, it is tempting to speculate that puncta formation and LLPS of 53BP1 at heterochromatin might be a dynamic process that constantly evolves or form during the cell division cycle. Similarly, BRCA1, the antagonizing factor for 53BP1 in DSB repair, was reported to co-localize with the XIST RNA in only 5–10% of cells[62] or partially (<40%) with centromeric heterochromatin[63] in a way very similar to 53BP1 puncta formation at heterochromatin; yet, it demonstrates significant biological functions in mitosis and genome stability. The published results and our data suggest that even a partial co-localization of

53BP1 puncta with heterochromatin is significant. In addition, the sometimes-close contact between 53BP1 puncta and HP1α indicates the presence of yet-to-be identified factors bridging these two proteins for the LLPS process at heterochromatin.

In our in vitro liquid droplet formation analysis, while we observed solid condensates, we also noticed that 53BP1-C1 and HP1α could form hollow sphere-like liquid droplets. Classically, liquid droplets are considered to be solid condensates, especially in in vitro assays. However, increasing number of studies showed vesicle-like liquid droplets formed by, for instance, germ granules of Drosophila[64,65], transcription factor TDP-43[66], and RNA-RNPs[67,68]. Such vesicle-like liquid droplets were considered to be formed through anisotropic protein–protein or protein–nuclei acid complexes that were probably driven by heterotypic electrostatic interactions between molecules. Hence, we speculate that the observed hollow structure of liquid droplets were also formed in a similar way by 53BP1 and HP1α. Our data provided support for this hypothesis. First, the formation of the vesicle-like liquid droplets depended on the concentrations of 53BP1 and HP1α used. At lower concentrations, the liquid droplets formed by 53BP1 and HP1α were largely small solid circles, whereas at higher concentrations, they formed both vesicle-like and solid droplets, which is consistent with the anisotropic interactions seen in RNA-RNPs[68]. Second, this could represent a unique feature for the pair of 53BP1 and HP1α in LLPS. Of note, these vesicle-like liquid droplets were relatively large in size (with a diameter of nearly 20 μm), indicating enhanced liquid surface tension in these droplets. Third, it remains to be determined if the addition of nucleosomal components would change the liquid droplet formation by 53BP1 and HP1α, which will be explored in detail in future studies.

Recently, it was suggested that 53BP1 formed liquid droplets at DSB foci[49,50], which also required the OD[49], similar to our observation. However, our studies differ from those mainly in the concept as to whether 53BP1's LLPS is related to DSB repair. Here, we presented evidence to show that 53BP1's LLPS is distinct from DSB repair, at least at heterochromatin. In conducting in vitro LLPS analysis, crowding agents known to enhance LLPS were used previously[49], but not in our assay, which led to different conclusions (53BP1 alone formed LLPS in vitro[49], but not here). We then presented a large amount of data to show that 53BP1 puncta involved different domains and residues from those required for DSB repair. Further, we characterized separation-of-function mutants of 53BP1 in DSB repair and LLPS, respectively, which cannot be explained if these 53BP1 puncta were also DSB foci. Overall, we believe that our studies have revealed a previously uncharacterized layer of regulation of 53BP1 in genome stability maintenance, which is different from its canonical role in DSB repair. Nevertheless, together with recent publications, they introduced the LLPS concept to 53BP1, broadening our understanding about 53BP1's biological function.

## Methods

**Cell cultures and transfection**. Parental or 53BP1 KO or KD MDA-MB-231, U-2 OS, MEF, HEK293T, IMR90 and APRE19 cells were cultured in DMEM with 10% FBS and 1% penicillin/streptomycin (HyClone). MCF-10A cells were cultured in DMEM:F12 (1:1) (HyClone, Thermo Fisher Scientific, Waltham, MA, USA) supplied with 1% penicillin/streptomycin (HyClone), 5% horse serum (HyClone), 0.1% insulin solution (Millipore/Sigma, St. Louis, MO, USA), 0.01% Cholera Toxin (Millipore/Sigma), 0.025% hydrocortisone (ACROS/Thermo Fisher) and 0.02% Epidermal Growth Factor (Life Technologies/Invitrogen, Carlsbad, CA, USA). Cells were maintained at 37 °C in 5% $CO_2$ and 98% humidity.

Transfection of plasmids was done with the X-tremeGENE HP DNA transfection reagent (#6366244001) from Millipore/Sigma (St. Louis, MO, USA). On the other hand, transfection of HEK293T cells to produce shRNA lentivirus was carried out using standard $Ca^{2+}$ transfection.

**Reagents and antibodies**. Rat monoclonal anti-53BP1 (#933002, clone W17184B, lot #B295535, mainly for IF) antibody was purchased from Biolegend (San Diego, CA, USA). Anti-H3K9me3 (#07-442), anti-p-histone H2A.X (Ser139) (for staining, JBW301, #05-636-MI), anti-p-HH3 (Ser10, #06-570), anti-RAD51 (#PC-130), anti-Cyclin A (#C4710), anti-53BP1 (MAB3802), anti-SC35 (#S4045) and anti-H2AK15ub (#MABE1119) antibodies were from Millipore/Sigma. Anti-HA (C29F4, #3724S), anti-HP1α (#2616S), anti-HP1β (#8676S) and anti-SUV39H1 (#8729) antibodies were from Cell Signaling Technology (Danvers, MA, USA). Mouse anti-HP1α (GA-62, #SC-130446), anti-β-Actin (C4, #SC-47778), anti-UHRF1 (H8) and anti-HP1γ (sc-398562) were from Santa Cruz Biotechnology (Santa Cruz, CA, USA). Anti-53BP1 (#NB100-304 and #NBP2-25028), anti-RIF1 (for staining, # NB100-1587), anti-TRF2 (#NB100-56506) and anti-GFP (#NB100-1770) antibodies were from Novus Biologicals (Centennial, CO, USA). Anti-ATM (2C1, #GTX70103) was from GeneTex Inc. (Irvine, CA, USA). Anti-pS824-KAP1 (#ab70369) was purchased from Abcam (Cambridge, UK).

Alexa Fluor conjugated secondary antibodies were purchased from Thermo/Invitrogen: Chicken anti-mouse IgG (H + L) Alexa Fluor 647 (#A-214463), Chicken anti-rabbit IgG (H + L) Alexa Fluor 647 (#A-21443), Donkey anti-mouse IgG (H + L) Alexa Fluor 568 (#A-10037), Donkey anti-rabbit IgG (H + L) Alexa Fluor 568 (#A-10042), Donkey anti-mouse IgG (H + L) Alexa Fluor 488 (#A-21202), Donkey anti-rabbit IgG (H + L) Alexa Fluor 488 (#A-21206), Donkey anti-rat IgG (H + L) Alexa Fluor 488 (#A-21208), Chicken anti-rat IgG (H + L) Alexa Fluor 488 (#A-21470).

Bleomycin (#AAJ60727MA) was from Alfa Aesare (Haverhill, MA, USA). ATM inhibitor KU-55933 (#S1092) was purchased from Selleckchem (Pittsburgh, PA, USA). Protein A (SC-2001) and Protein G PLUS agarose (SC-2002) were from Santa Cruz. Human IL-6 (#430504) and IL-8 (#430501) ELISA MAX Deluxe kits were purchased from Biolegend (San Diego, CA, USA).

**Plasmid construction**. To generate GFP-53BP1 FL and mutant constructs, Myc-tagged full-length human 53BP1 (a gift from Dr. Junjie Chen, MD Anderson) was used as the template to perform PCR and the resulting PCR products were cloned into the XhoI and KpnI sites of pEGFP-C1. Mutagenesis (e.g., FL/D1521R, C1/D1521R, and C1/OD-4A) was carried out using the QuickChange II site-directed mutagenesis kit (#200518) from Agilent Technologies (Santa Clara, CA, USA). The four conserved residues in the OD were firstly mutated in a pMX-53BP1-DB-HA vector, which lacks the two C-terminal BRCT domains. To generate the GFP-53BP1-FL/OD-4A mutant, the pMX-53BP1-DB-HA/OD-4A construct was digested with BlpI and SgrAI to release the residues 917-1751 of 53BP1. The corresponding domain in the GFP-53BP1-FL/WT was then replaced with the fragment from the pMX-53BP1-DB-HA/OD-4A vector to produce GF-53BP1-FL/OD-4A. All constructs were confirmed by DNA sequencing.

**shRNA transduction for generation of stable U-2 OS, HepG2, and HEK293T cell lines**. HEK293T cells seeded in 6-well plates were co-transfected with lentivirus vectors targeting 53BP1, HP1α, HP1β or HP1γ (1 μg), pMDL (0.65 μg), VSVG (0.35 μg) and RSV-REV (0.25 μg) to produce lentiviral particles using $Ca^{2+}$ transfection. After 48 h of transfection, virus-containing media was collected by centrifugation at ~1000 × g for 10 min. For target cell infection, equal amount (volume) of virus particles from independent shRNA was mixed and mixed with fresh media at a 1:1 ratio (viral media: fresh growth media). The mixed media were added into target cells plated one day prior to infection in the presence of polybrene (4 μg/ml). After 72 h of incubation, virus-infected cells were trypsinized and transferred to 10 cm dish for selection of stable clones in the presence of puromycin (1 μg/ml). Lentiviral shRNA vectors targeting human genes were purchased from Millipore/Sigma (#TRCN0000018865 and TRCN0000018869 for 53BP1; TRCN00002379990, TRCN0000344645, and TRCN0000344646 for HP1α; TRCN0000062223 and TRCN0000062224 for HP1β; TRCN0000021917 and TRCN0000021916 for HP1γ).

**Immunofluorescence staining, confocal imaging, and data analysis**. Cells grown on glass coverslips were transfected or not with GFP-tagged 53BP1 constructs, treated or not with 5 Gy IR or 3.5 μM bleomycin for 6 h (or as indicated in specific experiments). Coverslips were gently washed with PBS and fixed in 3.7% (vol/vol) formaldehyde for 10 min. After fixing, the cells were quenched by washing two times in 0.1 M glycine in PBS for 10 min. For the rat monoclonal anti-53BP1 antibody, an additional methanol treatment step was included before the cells were permeabilized and blocked with 10% (vol/vol) FBS and 0.4% Triton X-100 in PBS for 20 min at room temperature. The coverslips were washed three times with washing buffer (PBS containing 0.2% Triton-100, 0.1% BSA) and incubated with primary antibodies in PBS containing 0.1% Triton X-100 at 4 °C overnight or at room temperature for 2 h (anti-γH2AX, 1:300; anti-53BP1, 1:300; anti-CS35, 1:2000; anti-TRF2, 1:500; anti-H3K9me3, 1:300; anti-HP1γ, 1:1000; anti-HP1β, 1:1000; anti-RIF1, 1:300; anti-HP1α, 1:300; anti-SUV39H1, 1:100; anti-Cyclin A, 1:300). The coverslips were washed three times with washing buffer and incubated with secondary antibodies (Alexa Fluor 488, 568, or 647; 1:1000) in PBS containing 0.1% Triton X-100 for 1 h in dark at room temperature, followed by washing three times for 10 min each. The coverslips were placed inversely onto glass slides mounted with the Prolong Gold Anti-fade with DAPI (Life Technologies/Invitrogen) and stored at 4 °C overnight.

Except indicated in a few cases where projection images were acquired using an inverted Leica laser microscope, the rest are single z-plane confocal images taken under a HC PL APO CS2 63x/1.40 oil objective lens by the Leica SP8 HyVolution coupled super-resolution confocal microscopy with air-cushion stabilizing system, which allows to capture images with up to 120 nm resolution. Multi-channel sequential imaging was taken with the following laser settings and detectors: 405 Diode (blue) by PMT1 (420–480 nm), Argon (20% laser power, green) by HyD2 (505–545 nm), HeNe 594 (red) by HyD 4 (590–627 nm) and HeNe 633 (Cyan) by HyD 5 (660–700 nm). The scan speed was 400 Hz. The pinhole was 95.6 μm and the pinhole Airy was 1 AU. The laser intensity, filter, and image size (2048 × 2048) were kept the same for all confocal experiments. The zoom factor was kept at 1 (i.e., no zoom in or out) for all experiments. The gain value for each channel was adjusted each time based on the staining conditions but maintained the same for each particular experiment.

Images for each channel (blue, green, red, and cyan) were exported as individual non-compressed TIFF files by the built-in LAS X version software and individually analyzed by Photoshop (version 21.2.12 and 23.0.2). For image analysis, only Brightness/Contrast of the entire image was adjusted manually, and the parameters were kept the same for the same batch of images under the same channel (different channels may be adjusted differently). Image merging and cropping were done in Photoshop. Scale bar was applied to individual images before cropping. The puncta size for 53BP1 (wild type or overexpressed mutants) and heterochromatin centers (DAPI staining) were analyzed by the NIH Image J software (version 1.52q) based on standard procedures. To preclude experimental variations, we only analyzed the puncta for 53BP1 or heterochromatin factors (DAPI in MEF or H3K9me3 in human or mouse cells) in images obtained from the same experiments that have been applied with the same Brightness/Contrast adjustment.

**Immunoblotting**. Cells were harvested by trypsinization, followed by centrifugation at 150 × g for 5 min in a tabletop centrifuge. Cell pellets were washed with PBS once and lysed in NP-40 lysis buffer (100 mM Tris-HCl, pH 7.6, 150 mM NaCl, 1% NP-40, and protease and phosphatase inhibitors including 1 mM $Na_3VO_4$, 1 mM PMSF, 1 mM DTT, 10 μg/ml aprotinin, 1 μg/ml leupeptin, 1 μg/ml pepstein, and 10 mM sodium fluoride) on ice for 30 min. Structure-bound proteins were extracted by sonication using three pulses, each 10 s on/off at 1% amplitude. Following sonication, samples were centrifuged at 13,000 × g for 10 min at 4 °C. Supernatant was collected, quantitated, and boiled with Lameilli sample buffer for 5 min, ran on a gradient (15%-10%-6%) SDS-polyacrylamide gel and transferred to PVDF membranes. The membranes were blocked in 5% non-fat milk or BSA in 1X TBST for 1 h, washed three times with 1X TBST for 10 min, incubated with primary antibodies (1:1000) at 4 °C overnight, washed in 1× TBST for 10 min 5 times, and incubated in secondary antibodies conjugated with HRP in 1× TBST for 1 h at room temperature. The membranes were washed in 1× TBST for 10 min 5 times, reacted with ECL solution and visualized by the Tanon 5200 Imager system (Tanon, Shanghai, China).

**Time-lapse microscopy**. U-2 OS cells were seeded at a density of $1 \times 10^5$ per well in a 35 mm dish with a #1.5 circular glass cover at the bottom. After 20 h, cells were transfected with indicated constructs. Live cell imaging was performed at various time points after transfection using a Leica DMI6000 inverted fluorescence microscopy with temperature-controlled heating system. Cells with GFP-53BP1 signal were imaged with the 20× objective lens, adjusted the focus and determined the exposure time and gain value based on the protein expression levels, marked the field with the built-in software, and acquired images every 5 min for 24–48 h.

**Liquid droplet treatment**. For the 1,6-hexanediol treatment experiments, MDA-MB-231 or U-2 OS cells were seeded in a six-well plate with multiple glass covers in each well for 24 h, transfected (U-2 OS) or not (MDA-MB-231) with GFP-53BP1-C1 for 48 h, treated with 5% 1,6-hexanediol for 5-10 min. The cells were visualized in live (U-2 OS cells) or the glass covers (MDA-MB-231 cells) were fixed immediately with 3.7% formaldehyde and processed for immunofluorescence staining for endogenous 53BP1 proteins where DAPI was used to counter stain the nuclei.

**Fluorescence recovery after photobleaching (FRAP)**. For FRAP analysis, cells were seeded at a density of $1 \times 10^5$ per well in a 35 mm dish with a #1.5 circular glass cover at the bottom. After 20 h, cells were transfected with various constructs based on specific experimental design for 48 h. FRAP assay was conducted on a heated (37 °C) stage of the Leica TCS SP8 Hyvolution confocal microscopy system using the Lasos LGK 7872 ML05 laser conjugated with the HC PL APO CS2 40x/1.30 oil objective lens for optimum resolution for bleaching. The scan speed was 400 Hz. The pinhole size was 65.3 μm and the pinholeAiry was 1 AU. The Argon laser was used for bleaching at a power of 79.3651 W. For qualitative experiments, the size and depth of points-of-interest were determined in snapshots of cells under ideal optical parameters. Fluorescence within the outlined circle was measured by a low laser power (0.5–10%) for up to 30 s at 1-s interval before the bleach (pre-bleach) and then photobleached with 20–100% laser power for 1 s depending on the initial fluorescence intensity of the interest spot. Fluorescence recovery was monitored by scanning the whole cell by low laser power (0.5–10% power) for 120–300 s at a 10-s interval to allow the intensity to reach a steady plateau. The scanning laser intensity did not significantly photobleach the fluorescence of specimens during the time course of the experiment. For calculation, the average

fluorescent intensity in the bleached areas was adjusted by subtracting background intensities in three unbleached spots in the same image. The resultant fluorescence intensities were normalized to the average of that of pre-bleach and then plotted over time (sec).

**ChIP-seq analysis of co-localization peaks**. The ChIP-seq data for 53BP1, γH2AX, H3K9me3, and H3K4me2/3 were downloaded from ArrayExpress (accession number: E-MTAB-5817)[40]. Raw sequencing reads were trimmed with Trim Galore v0.6.4 (https://www.bioinformatics.babraham.ac.uk/projects/trim_galore/) and aligned to the human reference genome (hg38) using Bowtie v1.3.0[69], followed by filtering for uniquely mapped reads. Duplicated reads were removed by Picard v1.119 (http://broadinstitute.github.io/picard/). ChIP-seq peaks was called using MACS2 v2.2.7.1[70] with -q 0.05. Peaks of 53BP1, H3K9me3, and H3K4me2 intersected with γH2AX were filtered to avoid the effect of DNA double-strand breaks. Then the remaining 53BP1 peaks overlapping H3K9me3 and H3K4me2/3 were obtained with BEDTools v2.28.0[71]. For Heat map analysis, ChIP-Seq bigwig files were generated using deeptools v2.5.3[72] bamCoverage function. ChIP-Seq heatmaps were generated with the deeptools computeMatrix and plotHeatmap functions to compare the enrichment between 53BP1 and H3K9me3.

**Expression and purification of proteins**. All proteins were expressed in *Escherichia coli* BL21 (DE3) cells. For expression of mCherry-HP1α, cDNA encoding Human HP1α was cloned into the pET-28a vector, incorporating a cDNA encoding mCherry with a seven-amino–acid linker (GSAAAGS) between HP1α and mCherry. For expression of mGFP-53BP1-C1, -C2, and -C3, the GFP-53BP1-C1 vector was used as the template to run PCR and the resultant PCR products for C1, C2, and C3 were cloned into pET-28a. The cDNA encoding mEGFP (Addgene plasmid 18696) followed by a GSAAAGS linker sequence was inserted into the N terminus of 53BP1-C1/C2/C3 in the pET-28a backbone.

All expression constructs were transformed into *E. coli* cells individually. Cells were grown at 37 °C in LB medium containing 50 µg/mL kanamycin until the OD reached 0.6. Cells were then induced with 0.5 mM isopropyl β-D-1-thiogalactopyranoside (IPTG) over night at 18 °C. Cells were harvested by centrifugation and resuspended in lysis buffer (50 mM NaH$_2$PO$_4$, 300 mM NaCl, 10 mM imidazole, 0.5 mg/ml lysozyme, and protease inhibitors, pH 8.0). After sonication, cell extracts were centrifuged at $25,000 \times g$ for 20 min, and the supernatants were bound to Ni Sepharose (GE, USA) and washed three times with washing buffer (50 mM NaH$_2$PO$_4$, 300 mM NaCl, 20 mM imidazole, pH 8.0). The proteins were eluted with elution buffer (50 mM NaH$_2$PO$_4$, 300 mM NaCl, 300 mM imidazole, pH 8.0) and protein was purified by dialysis and concentrated using Amicon Ultra-4 centrifugal filter units. All purified proteins were quantified by the Bradford assay.

**In vitro liquid–liquid phase separation assay**. In vitro liquid phase separation assay was performed using phase buffer (20 mM HEPES pH 7.4, 100 mM NaCl, 1 mM DTT). Briefly, mCherry-HP1α (0, 0.1, 0.25, 1, and 10 µM) was mixed or not with 53BP1-C1/C2/C3-GFP (0, 0.1, 0.25, 1, and 10 µM) in a 0.2 ml Eppendorf tube on ice and then incubated at room temperature for 20 min. After incubation, 5 µl mixture were spotted on a glass slide and sealed with coverslips. The slides were then imaged with an Olympus IX73 epifluorescence microscope.

**Comet assay**. We performed the alkaline comet assay as we previously reported with modifications[73] using the Cell Biolabs kit (#STA-351, San Diego, USA). Briefly, parental or engineered U-2 OS cells were treated or not with 3.5 µM bleomycin for 6 h, collected by trypsinization, washed once with PBS and then re-suspended in PBS at a final cell concentration of $10^5$ cells/mL. Ten µl of cell suspension were mixed with 100 µl low melting agarose, which were poured onto the supplied glass slides. Cell lysis was carried out at 4 °C overnight and electrophoresis was done in a tank containing the alkaline running buffer (1 mmol/L EDTA, 300 mmol/L NaOH, pH 13) at a voltage of 20 volts (300 mA) for 20 min. The slides were dried and stained with the supplied Vista Green DNA Dye for 10 min according to the manufacturer's instructions. Images were acquired using an inverted fluorescence microscopy and analyzed using the CaspLab software[74].

**ELISA analysis for IL-6 and IL-8**. Secretory IL-6 (#BD550799) and IL-8 (#BD550999) levels were measured using the ELISA kit purchased from BD (Franklin Lakes, NJ, US). Briefly, U-2 OS parental, 53BP1 KO or KO cells reconstituted with different GFP-53BP1 constructs were treated with 0.35 µM bleomycin for 7 days. The cell culture conditional media were collected and centrifuged at $13,000 \times g$ for 10 min at 4 °C. ELISA was performed following the manufacturer's protocol. The absorbance was normalized by cell number in each group and the level of IL6 and IL8 was determined by the standard curve.

**ChIP-qPCR and qPCR**. $1 \times 10^7$ U-2 OS, MEF or HEK293T parental, 53BP1 KO, KD or GFP-53BP1-FL/WT reconstituted cells were subjected to ChIP using anti-53BP1 or anti-GFP antibodies (1 µg antibody per 5 µg isolated chromatin DNA was used for all ChIP experiments). Immunoprecipitated DNA were column-purified

and subjected to quantitative PCR (qPCR) analysis on a QuantStudio 3 Real-Time PCR System. Primers used are as follows:
SATIII F (5′-3′): AATCAACCCGAGTGCAATCGAATGAATG
SATIII R (5′-3′): TCCATTCCATTCCTGTACTCGG
mcBox F (5′-3′): AGGGAATGTCTTCCCCATAAAAACT
mcBox R (5′-3′): GTCTACCTTTTATTTGAATTCCCG
SATα-F (5′-3′): CTCACAGAGTTGAACGATCCT
SATα-R (5′-3′): ATTCTACCATTGACCTCAAAGCG
CH21-F (5′-3′): GTCTACCTTTTATTTGAATTCCCG
CH21-R (5′-3′): AGGGAATGTCTTCCCCATAAAAACT
HPRT1-F (5′-3′): AGCTTGCTGGTGAAAAGGA
HPRT1-R (5′-3′): CCAAACTCAACTTGAACTCTCATC
MajSAT-F (5′-3′): GGCGAGAAAACTGAAAATCACG
MajSAT-R (5′-3′): CTTGCCATATTCCACGTCCT
MinSAT-F (5′-3′): TTGGAAACGGGATTTGTAGA
MinSAT-R (5′-3′): CGGTTTCCAACATATGTGTTTT
β-actin F (5′-3′): GTCCCTCACCCTCCCAAAAGC
β-actin R (5′-3′): GCTGCCTCAACACCTCAACCC

**Statistics and reproducibility**. Immunoblotting and immunostaining experiments were performed at least two times that produced similar results. Data are presented as mean values ± standard deviation (SD). While the statistical analysis for the correlation between Cyclin A expression levels and the number of 53BP1 puncta was performed by Pearson's Correlation calculation (https://www.socscistatistics.com/tests/pearson/default2.aspx), the rest was conducted by Prism 9.0. For two groups, two-tailed unpaired *t*-test was used to determine statistical difference. For more than two group comparison, Ordinary one-way ANOVA test was conducted. *P*-values of less than at least 0.05 were considered statistically significant.

**Reporting summary**. Further information on research design is available in the Nature Research Reporting Summary linked to this article.

## Data availability
The data that support this study are available from the corresponding authors upon reasonable request. ChIP-seq data for 53BP1 were downloaded from GEO database (accession number: GSE108114) or E-MTAB-5817 from EMBL-EBI ArrayExpress database (accession numbers: ERR2008219, ERR2720661, and ERR2720666) and SRR10540101 from SRA. All sequencing reads were trimmed using Trim Galore v0.6.5 (http://www.bioinformatics.babraham.ac.uk/projects/trim_galore/). Reads were mapped to hg38 version of human reference genome using Bowtie2 v2.4.1 allowing only unique alignment[75]. PCR duplicates were removed using Picard v1.114 (http://broadinstitute.github.io/picard). Peaks were called using MACS2 v2.1.1 with -p 0.05[70]. Additional ChIP-seq data for H3K9me3 and H3K27ac were downloaded from ENCODE (https://www.encodeproject.org/). Source data are provided with this paper.

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

## Acknowledgements

We thank Kuntian Luo, Zhenkun Lou, Neil Johnson, Zihua Gong, Linyu Lu, Daniel Durocher, and Junjie Chen for providing cell lines or 53BP1 plasmids. We thank John Pink for reading the manuscript. The authors also thank Mike Sramkoski and Richard Lee for microscopy assistance. Y.X. is supported by NIH R01 GM110132. This study was supported by the American Cancer Society (RSG-15-042 DMC), NIH (R01CA230453), and Case Cancer Center pilot grant (CA043703) to Y. Zhang. The Cytometry & Imaging Microscopy Shared Resource is supported by the Case Cancer Center grant (P30CA043703). The Medical School Microscopy Core is supported by the Shared Instrumentation Grant S10 OD024996 from NIH.

## Author contributions

L.Z. and Y.Z. conceived the project. L.Z., X.G. and F.W. contributed equally to this study. J.T., I.Y., A.S., S.J., M.C., M.T., F.P., W.W., J.W., X.G., M.M., F.J., and X.Y. participated in mutagenesis experiments, imaging acquisition, data analysis, and discussion. M.W. and Y.X. contributed to yeast analysis. Y.Z. wrote the manuscript.

## Competing interests

The authors declare no competing interests.
