## [Peer Review File · Nature Communications]

53BP1 regulates heterochromatin through liquid phase separationEditorial Note: This manuscript has been previously reviewed at another journal that is not operating a transparent peer review scheme. This document only contains reviewer comments and rebuttal letters for versions considered at Nature Communications. Mentions of the other journal have been redacted.

REVIEWER COMMENTS

Reviewer #1 (Remarks to the Author):

While I acknowledge the extensive efforts made by the authors to strengthen their work, a remaining weakness of this study is that it remains largely unclear how 53BP1 contributes to heterochromatin integrity and genome stability maintenance via puncta formation involving LLPS. However, given that editorially the mechanistic insights are considered sufficient for Nature Communications, the manuscript may now possibly be acceptable overall for publication, despite I maintain my reservations. Apart from the lack of mechanistic insights, my major outstanding concern relates to the way the authors defined and quantified 53BP1 puncta, which to this reviewer at least remains highly confusing and seems to lack rigor, especially given that different criteria were applied to define puncta for different 53BP1 constructs: To discriminate between 53BP1 DSB repair foci and puncta in undamaged cells, the authors considered 1-3 nuclear 53BP1 dots per cell to be spontaneous DSB repair foci, whereas 4 or more dots per cell were counted as puncta. This seems like a rather arbitrary definition, and it would be much more appropriate to simply use colocalization (or the lack thereof) with a DNA damage marker such as γ H2AX (e.g. as in Fig. S8 and S9) as a reliable criterion for whether the 53BP1 dots correspond to DSB repair foci or puncta. In cells exposed to DNA damage, the described criteria used were even more vague and confusing: for instance, for endogenous or overexpressed full-length wild-type 53BP1, dots were considered to be DSB repair foci as 'the puncta look like foci', whereas for 53BP1-C1 constructs the distinction between DSB repair foci and puncta was apparently made on the basis of the dots being distributed throughout the nucleus (puncta) or at certain regions of the nucleus (DSB repair foci), as well as their overall size (puncta being often larger in size than DSB repair foci). Based on these considerations the authors then stated that they used an arbitrary cutoff of >10 dots as a criterion for DSB repair foci, even though it appears that more than 10 53BP1 puncta per cell are often formed. This renders some results confusing and difficult to interpret. For instance, does the data on puncta/DSB repair foci formation of the 53BP1 C1-DR mutant in Fig. 6c,d (puncta formation in 40% of undamaged cells, no formation of DSB repair foci in cells exposed to DNA damage) imply that the formation of dots (puncta?) is unaffected by DNA damage or that 53BP1 C1-DR puncta are disassembled upon DNA damage? Again, in the absence of specific markers that can reliably classify the 53BP1 dots as puncta, it may perhaps be preferable to apply the lack of colocalization with DNA damage markers (e.g. as in Fig. 7d) as a more rigorous criterion for 53BP1 puncta that do not correspond to DSB repair foci. In my opinion, it will be important to clarify this issue prior to publication in order to avoid confusion.

Reviewer #4 (Remarks to the Author):

Zhang and colleagues identify a novel role for the 53BP1 protein, a key regulator of double-stranded DNA repair, in heterochromatin maintenance by promoting liquid-liquid phase separation. In a series of experiments performed with multiple mutants of 53BP1 in a wide selection of distinct cellular contexts, the authors show that 53BP1 forms distinct nuclear foci related to the different functions of the protein in DNA repair and heterochromatin maintenance.

Overall, the manuscript presents an elegant dissection and identification of the domains and residues of 53BP1 involved in a previously unknown LLPS-formation at heterochromatin. In particular, the variety of 53BP1 mutant constructs and the wide panel of cell-lines tested provide strong confidence in this manuscript. However, while the authors addressed most concerns raised by reviewers at Nature Cell Biology, I agree with Reviewer #1 and #3 that data presented still does not address how 53BP1 supports heterochromatin integrity via LLPS and importantly that this novel role contributes to genome stability by heterochromatin integrity. Authors still do not present data demonstrating that relaxation of heterochromatin leads to genome instability (aneuploidy, mitotic arrest, DNA damage

foci, etc). This is important as the bleomycin double-strand induction experiments presented in Figure 8, whilst informative for the separation of function, do not directly address the impact of 53BP1-LLPS loss at heterochromatin.

I also agree with reviewer #1 that the co-IP experiments in S13 are still not conclusive and this experiment should be revisited as it is so crucial for the mechanistically novelty of this manuscript. Similarly, I also query the distribution of 53BP1 and heterochromatin (please see more details below). Whilst the localization of 53BP1 at heterochromatin is convincing, the data can also support an independence between 53BP1-LLPS and HP1a-LLPS puncta. Further, dislodgement of 53BP1 from heterochromatin upon HP1a downregulation (Fig S12b) could simply be an indirect consequence of relaxation of heterochromatin, rather than a direct effect of 53BP1-HP1a interaction.

Additionally, the manuscript is largely requiring methodology details, rendering the ability to confirm some of the author's claims with the data presented. For example, "puncta rate" is widely used to characterize distinct 53BP1 foci and their function but there is no explanation on how 53BP1 foci/puncta are defined besides from a statement about the differences in number of foci (<3 or >4). The manuscript would be greatly improved by the addition of detailed information about methods, in particular imaging analysis, both in the text and in the figures to enable an appropriate interpretation of the data and experimental procedures by the reader. Specific comments are detailed below.

Major comments:

1. Detailed description of methodology, in particular of immunofluorescence imaging analysis is lacking. "Puncta rate" definition has been included in lines 103-104 but information as to how the foci are defined (p.e. masking/thresholding manually or automatically, software used, etc) should be defined. Also, are the images shown projections or single-lanes? This information should be properly provided in the manuscript.
2. Similarly, the number of foci/puncta per condition would be more informative than an arbitrary number to define puncta versus puncta. The authors should include histogram of densities to show the variation in number of 53BP1 (or H3K9me3) foci to strengthen a functional difference associated to 53BP1 foci number. This is crucial as most of the claims stem from the difference between puncta/foci and both previous reviewers and myself find the definitions provided quite insufficient.
3. The statistical information is also poorly described through the manuscript. Often the number of cells or foci analysed are not mentioned in the text, figures nor in the legends and when "n" is mentioned, there is often discrepancy to what data shows. For example, several figure legends mention that each dot represents a cell but the number of dots often does not match the "n" provided (for example S2B n= 425 for parental but there are only 37 dots).
4. Overall, the organization of the supplementary figures is very confusing. There are many supplementary figures for each main figure and a logic order following the mentions within the manuscript is not followed. For example, in the second paragraph of first result section (line 129-136), text refers to s8 and figure 7 (before s2-4 are even mentioned in the text). This back and forth organization is very distracting to the reader, further rendering proper interpretation of data. Strongly recommend authors to re-organize and merge the supplementary Figures.
5. Often, the DAPI staining in the representative images selected for the parental example shows features of unhealthy cells (p.e. micronuclei or apoptotic-associated DAPI staining is visible in S3a, S9a, S9b, S10a, S10f). The authors should perform apoptosis analysis to demonstrate that the large 53BP1 foci numbers observed in their parental cell-lines are not resulting from stress during culturing conditions. This would further support that this puncta are spontaneous.
6. The claim that 53BP1 is required for heterochromatin formation is not substantiated by the data presented through the manuscript and therefore should be removed.

7. Finally, authors should discuss if 53BP1 puncta inhibit repair-associated 53BP1 foci formation as puncta seem to occur concurrently with replication (cyclin A positive cells).

Minor comments:

9. Puncta size and intensity (of both 53BP1 and heterochromatin) are mentioned through the manuscript without any analysis of such features described or shown (p.e. lines 148-149, 216, 270-271). Please perform the analysis and include description of methods used both in legends and methods section.

10. Figure 2: Related to reviewer #1 comment about 53BP1 location at heterochromatin, how were images acquired in different channels? 53BP1 staining seems to always border the left side of the heterochromatin-foci (H3K9me3 signal), which could be a consequence of plate-drift during imaging (if sequential acquisition was performed).

11. Figure 2i: the y-axis defines #DAPI centers/cell but the interpretation of these results in the text mentions size differences, rather than number differences: "reduction in the number of large heterochromatin centers accompanied with an increase in smaller ones" (lines 148-149 and later in line 216). What is this panel showing? Please correct accordingly and add information on how analysis was performed, including how DAPI centers were defined.

12. Figure 2f&g: There are 40-50% of cells in the parental line with less than 3 H3K9me3 foci. This result is quite surprising as H3K9me3-foci are quite prevalent in all cell-types. Details on how H3K9me3-foci were defined need to be provided to enable the interpretation of an otherwise unexpected result.

13. Figure 4: Please define mcBox, SatIII and ALR to allow for a better understanding of the heterochromatic loci probed, similarly to the description provided to murine major and minor satellites.

14. Figure S15a does not show that "53BP1-C1 puncta completely overlapped with HP1a or H3K9me3", as stated in line 251. There is a better overlap between marks but many HP1a or H3K9me3 foci are not overlapping with 53BP1-C1. Please correct statement.

15. Line 252-253: "These results suggest that 53BP1-C1 promoted heterochromatin formation". These results do not support this claim, but rather suggest that 53BP1 is recruited to heterochromatin. Please remove this phrase.

16. Figure S16a: The images chosen are oversaturated and the claim that puncta can be observed through the interphase and are resolved during mitosis, as claimed in the manuscript (line 275-276), cannot be confirmed. Please provide better images to support this statement. Also, times are typically shown as timepoints (t0, t2, t4, t10, etc) as the actual timing of acquisition is not informative.

17. Please explain the score in S16b in the legend.

18. For the 1,6-hexanediol treatment, why was it performed in fixed cells? Live imaging with the constructs GFP-53BP1-C1 and/or GFP-HP1a shown by the authors would be more adequate to show that these foci are sensitive to this alcohol. Further, as the authors use mCherry-HP1a, these experiments in cells co-expressing both constructs would additionally show whether 53BP1 and HP1a belong to the same LLPS foci or not.

19. Fig17a-c: these images are oversaturated making it difficult to interpret the same as the authors did. In particular, 53BP1-C1 doesn't seem to be affected by 1,6-hexanediol treatment, as puncta are still visible (particularly in S17c).

20. Figure S17d: Do cells survive after 5-10min of 1,6-hexanediol treatment? Most studies published in different cells (and organisms) show that this alcohol is toxic after 10min, leading to apoptosis so this result is very surprising. Please discuss.

21. Figure 5c: Why are the droplets formed hollow in the center? LLPS droplets are formed by separation of interacting molecules from the solution, which creates a homogeneously isolated foci. Thus, the empty core of the droplets shown needs to be discussed. Also, an experiment where increased concentrations of either HP1a or 53BP1-truncations would probe for their dependency in forming LLPS droplets.

22. Figure 5f: the images in HP1a KD panel at 60 and 120 seconds are confusing ... At 60 seconds the bottom highlighted chromocenter shows recovery of fluorescence but it is again lost at 120 seconds?

23. Figure 6: The number of cells analysed has not been included in the Figure or legend.

24. Figure 8f: Please define CH21 (as in comment 13 above).

25. Figure S13: ChIPseq analysis is very interesting but a genome wide analysis should be shown (for example showing the number of overlapping peaks). The 5 loci presented are not enough to claim that the overlap observed by immunofluorescence is corroborated by ChIPseq data.

26. Line 371-374: The blot presented to show that OD-4A interacts less with FLAG-HP1a is not very robust. The authors should replace for a more clearer (stronger GFP band), which would better allow to infer a lower interaction.

27. Line 390: The different definitions of the foci according to treatments seems very subjective. Again, a simple histogram density distribution of number of foci (automatedly defined) should unbiasedly show differences in foci formation.

28. Line 481: "reserve" should be "preserve".

29. Axis of puncta rate should be standardize to show 0-100, for consistency.

30. Please replace "heterochromatins" with "heterochromatin".

31. Line 168: "binging" should be "binding".

32. Line 227: please change phrase to "53BP1-FL/WT slightly reduced, although significantly, the elevated levels of heterochromatin" to better reflect data shown.

33. Line 249: Please change to "53BP1-C1 puncta elicited an increased number of DAPI foci" to more accurately reflect the data.

We thank the two reviewers (Reviewer #1 and #4) for the constructive critiques. We have revised the manuscript based on these critiques. Addressing these concerns has substantially improved the manuscript. Major changes are outlined below.

- We now used co-localization between 53BP1 ‘dots’ and foci of a DSB marker (γ H2AX, pKAP1 or RIF1) to discriminate between the puncta and DSB foci of 53BP1 as suggested by Reviewer #1 and #4. If the 53BP1 ‘dots’ largely co-localize with DSB foci of one of these markers, then these ‘dots’ are considered to be DSB foci and this cell will be counted as a DSB foci positive, but puncta/LLPS negative cell. However, if there is limited co-localization (including the number of co-localizing dots and the extent of co-localization between 53BP1 dots and DSB foci), then these 53BP1 ‘dots’ are counted as the LLPS-related puncta and accordingly these cells are scored as puncta positive (but DSB negative). We also measured the size of 53BP1-C1 puncta and DSB foci to show that the former is larger than the latter, helping us to distinguish puncta from DSB foci.
- We have performed a number of new experiments to address concerns raised by Reviewer #4, including (1) evaluation of cell death of **parental** cells forming 53BP1 puncta; (2) revision of co-IP between 53BP1 and HP1 proteins; (3) Heat map analysis of genome-wide co-localization between 53BP1 and H3K9me3; (4) determination of GFP-53BP1-C1 and mCherry-HP1 α puncta disappearance by 1,6-hexanediol in live cells; (5) analysis of in vitro liquid droplet formation by increasing the concentration of GFP-53BP1-C1 or mCherry-HP1 α .
- We have re-analyzed 53BP1 puncta distribution including both the number per cell and their size, which provided strong support to the previously demonstrated 53BP1 puncta rate data to address the Reviewer #4’s concern.
- We provided more details about the experimental procedures and the analysis of 53BP1 and heterochromatin puncta, as well as microscopy details.
- We also significantly re-organized and merged the supplementary figures. Despite the addition of more than 10 pieces of new data, we managed to reduce the number of supplementary figures from 19 to 15.
- Other concerns are addressed point-by-point below.

Changes are highlighted in blue in the revision.

REVIEWER COMMENTS

Reviewer #1 (Remarks to the Author):

While I acknowledge the extensive efforts made by the authors to strengthen their work, a remaining weakness of this study is that it remains largely unclear how 53BP1 contributes to heterochromatin integrity and genome stability maintenance via puncta formation involving LLPS. However, given that editorially the mechanistic insights are considered sufficient for Nature Communications, the manuscript may now possibly be acceptable overall for publication, despite I maintain my reservations.

Response: We first thank the reviewer for supporting the publication of our manuscript at Nature communications.

In my humble opinion, we have provided several lines evidence, including those newly added data to address the concerns from Reviewer #4, to show that the LLPS function of 53BP1 contributes to heterochromatin integrity and genome stability: (1) 53BP1 forms LLPS puncta at heterochromatin in a way dependent on the key heterochromatin protein HP1 α ; (2) in turn 53BP1 stabilized the retention and the mobility of HP1 α at heterochromatin, (3) loss of 53BP1 led to reduced structural integrity, de-repression of heterochromatin and increased DNA damage, and (4) re-constitution of LLPS proficient (regardless of DSB repair capability), but not LLPS deficient, 53BP1 mutants restored the heterochromatin structure, transcriptional repression and reduced the elevated DNA damage and senescence caused by DNA damage in 53BP1 KO or KD cells. Combined, we concluded that 53BP1's LLPS function contributes, at least partially, to heterochromatin function and genome stability maintenance, which is now discussed in the manuscript (lines 558-568).

Nonetheless, we understand the Reviewer's concern and have revised the discussion section accordingly. We stated that further investigation is needed to fully understand the significance of 53BP1's LLPS function in maintaining heterochromatin and genome stability in future studies (lines 588-590). In addition, we modified the Abstract to tune down our conclusion and perspectives (lines 38-41).

Apart from the lack of mechanistic insights, my major outstanding concern relates to the way the authors defined and quantified 53BP1 puncta, which to this reviewer at least remains highly confusing and seems to lack rigor, especially given that different criteria were applied to define puncta for different 53BP1 constructs: To discriminate between 53BP1 DSB repair foci and puncta in undamaged cells, the authors considered 1-3 nuclear 53BP1 dots per cell to be spontaneous DSB repair foci, whereas 4 or more dots per cell were counted as puncta. This seems like a rather arbitrary definition, and it would be much more appropriate to simply use colocalization (or the lack thereof) with a DNA damage marker such as γ H2AX (e.g. as in Fig. S8 and S9) as a reliable criterion for whether the 53BP1 dots correspond to DSB repair foci or puncta.

Response: *We thank the Reviewer for this great suggestion. As suggested by the Reviewer, we have revised the manuscript to discriminate 53BP1 puncta from DSB foci by asking whether they co-localize with one of the DSB markers (γ H2AX, pKAP1 or RIF1 foci). We also updated the way to show the data by also presenting puncta distribution (i.e., puncta number per cell) as suggested by Reviewer #4. By doing so, we can provide a straightforward illustration about the changes of 53BP1 puncta.*

Further, as asked by Reviewer #4, we analyzed the size of endogenous and exogenous 53BP1 puncta. For endogenous 53BP1 puncta, the median size was determined to be $\sim 1.243 \mu\text{m}^2$ (area) with the 25% and 95% percentiles to be 0.680 and $5.178 \mu\text{m}^2$, respectively (Fig. S1f). When we compared the size of GFP-53BP1-C1 puncta and DSB foci side-by-side, we determined the median, 25% and 75% percentiles of puncta to be 6.355 , 3.290 and $7.913 \mu\text{m}^2$ (in area), respectively, which were all larger than those for DSB foci at 0.377 , 0.274 and $0.592 \mu\text{m}^2$ (in area), respectively (Fig. S14e). These data suggest that the puncta had generally larger size than the DSB foci as we previously described (although not quantitatively).

Combined, we believe that now we have provided several unbiased yet quantifiable criteria in the revision to discriminate between 53BP1 puncta/LLPS and DSB foci (lines 95-97, 295-297, 312-324, 437, 452, and 469-472). We hope these will address the Reviewer's concern.

In cells exposed to DNA damage, the described criteria used were even more vague and confusing: for instance, for endogenous or overexpressed full-length wild-type 53BP1, dots were considered to be DSB repair foci as ‘the puncta look like foci’, whereas for 53BP1-C1 constructs the distinction between DSB repair foci and puncta was apparently made on the basis of the dots being distributed throughout the nucleus (puncta) or at certain regions of the nucleus (DSB repair foci), as well as their overall size (puncta being often larger in size than DSB repair foci). Based on these considerations the authors then stated that they used an arbitrary cutoff of >10 dots as a criterion for DSB repair foci, even though it appears that more than 10 53BP1 puncta per cell are often formed. This renders some results confusing and difficult to interpret. For instance, does the data on puncta/DSB repair foci formation of the 53BP1 C1-DR mutant in Fig. 6c,d (puncta formation in 40% of undamaged cells, no formation of DSB repair foci in cells exposed to DNA damage) imply that the formation of dots (puncta?) is unaffected by DNA damage or that 53BP1 C1-DR puncta are disassembled upon DNA damage? Again, in the absence of specific markers that can reliably classify the 53BP1 dots as puncta, it may perhaps be preferable to apply the lack of colocalization with DNA damage markers (e.g. as in Fig. 7d) as a more rigorous criterion for 53BP1 puncta that do not correspond to DSB repair foci. In my opinion, it will be important to clarify this issue prior to publication in order to avoid confusion.

Response: *We appreciate the Reviewer for pointing this and apologize not to clarify how DSB foci were determined in Fig. 6d. In fact, in this case (after IR or bleomycin treatment), we used co-localization with γ H2AX (as recommended by the Reviewer) to determine if C1 or C1-DR formed DSB foci, but we forgot to provide detailed explanation in the previous version of the manuscript. Exactly as suggested by the reviewer, in the presence of DNA damage, if the 53BP1 dots largely co-localized with those of γ H2AX, they were counted as DSB foci; however, if they rarely co-localized with γ H2AX foci, we considered them as 53BP1 puncta/LLPS. A representative image is shown in Supplementary Fig. S12b. We have revised the manuscript accordingly, which we believe can clear the confusion about the distinction between 53BP1 puncta and DSB foci (lines 312-324, 437, 452, and 466-469).*

As to the question about whether DNA damage would affect 53BP1 puncta formation, we have discussed this point in the previous revision, which was to answer a question from the previous Reviewer #3. In brief, DNA damage does not affect 53BP1's puncta formation as shown in Fig. S12c (line 318-320).

Reviewer #4 (Remarks to the Author):

Zhang and colleagues identify a novel role for the 53BP1 protein, a key regulator of double-stranded DNA repair, in heterochromatin maintenance by promoting liquid-liquid phase separation. In a series of experiments performed with multiple mutants of 53BP1 in a wide selection of distinct cellular contexts, the authors show that 53BP1 forms distinct nuclear foci related to the different functions of the protein in DNA repair and heterochromatin maintenance.

Overall, the manuscript presents an elegant dissection and identification of the domains and residues of 53BP1 involved in a previously unknown LLPS-formation at heterochromatin. In particular, the variety of 53BP1 mutant constructs and the wide panel of cell-lines tested provide strong confidence in this manuscript.

Response: *We thank the reviewer for the positive comments on our study.*

However, while the authors addressed most concerns raised by reviewers at Nature Cell Biology, I agree with Reviewer #1 and #3 that data presented still does not address how 53BP1 supports heterochromatin integrity via LLPS and importantly that this novel role contributes to genome stability by heterochromatin integrity. Authors still do not present data demonstrating that relaxation of heterochromatin leads to genome instability (aneuploidy, mitotic arrest, DNA damage foci, etc). This is important as the bleomycin double-strand induction experiments presented in Figure 8, whilst informative for the separation of function, do not directly address the impact of 53BP1-LLPS loss at heterochromatin.

Response: *In our humble opinion, we have provided orthogonal lines of evidence to support the role of 53BP1's LLPS in maintaining the heterochromatin integrity. These include: (1) 53BP1 forms LLPS puncta at heterochromatin in a way dependent on the key heterochromatin protein HP1 α ; (2) in turn 53BP1 stabilized the retention and the mobility of HP1 α at heterochromatin, (3) loss of 53BP1 led to reduced structural integrity, de-repression of heterochromatin and increased DNA damage, and (4) re-constitution of LLPS proficient (regardless of DSB repair capability), but not LLPS deficient, 53BP1 mutants restored the heterochromatin structure (heterochromatin center structure recovery in MEF and 53BP1 association with heterochromatin centers in HEK293T and U2OS) and transcriptional repression caused by 53BP1 deletion. These rescue experiments were conducted in MEF, HEK293T and U2OS cells with either 53BP1 knockout or knockdown. Hence, we concluded that 53BP1's LLPS function contributes to heterochromatin structure and function (lines 558-568).*

As to the role of 53BP1's LLPS in genome stability protection through heterochromatin, the Comet assay results and the ELISA data in Fig. 8 demonstrated a protective role of LLPS proficient 53BP1 proteins (DSB repair deficient) in reducing DNA damage and senescence induced by DNA damage in 53BP1 KO cells, which at least partially addressed the Reviewer's concern about the contribution of 53BP1's LLPS to genome stability in our opinion. In addition, it has been previously reported that 53bp1^{-/-} mouse embryonic fibroblasts growing under normal conditions tended to be aneuploid and tetraploid (Ward 2003), displaying chromosome segregation errors during mitosis, which is similar to cells with heterochromatin defects due to the loss of HP1 α (Peng 2009). These reports are also consistent with our model in which the LLPS function of 53BP1 facilitates the genome stability (lines 556-558).

Nonetheless, we agree with the Reviewer that whether the 53BP1's LLPS function protects genome stability through maintaining heterochromatin integrity needs to be further investigated. The following approaches can be considered to address this question. First, using the reconstituted U2OS cell lines expressing different 53BP1 constructs, we could introduce a specific cut at a pre-defined heterochromatin locus by CRISPR/Cas and then perform RNA-seq and DNA-seq (even ATAC-seq and/or Hi-C) to determine both the DNA structural integrity and transcriptional repression at heterochromatic repetitive regions. Additionally, we will examine genomic instability (aneuploidy, mitotic defects, DNA damage, etc., as suggested by the Reviewer) in this situation. Second, to determine the role of heterochromatin in such effect of 53BP1, we could deplete HP1 α in these cells to show that in the absence of HP1 α , the 53BP1-LLPS proficient mutant will not protect heterochromatin integrity, nor genome stability. Further, we could expression 53BP1-LLPS proficient but DSB repair deficient mutant to determine if such mutant could convert euchromatin into heterochromatin. We believe that these experiments should further prove the involvement of heterochromatin in the protective function of 53BP1-LLPS in genome stability.

Having said that, given the scope and conceptual advances that this manuscript already has, we hope that the Reviewer would agree with us that such kinds of experiments are best pursued in an independent study in the near future. Reviewer #1, who originally raised this concern, now agrees that the conceptual advance and innovation of this manuscript are acceptable for publication in Nature Communications.

Nevertheless, we have revised both the Abstract (lines 38-41) and the Discussion section (lines 558-568, 582-590) accordingly to reflect the concern of the Reviewer.

I also agree with reviewer #1 that the co-IP experiments in S13 are still not conclusive and this experiment should be revisited as it is so crucial for the mechanistically novelty of this manuscript. Similarly, I also query the distribution of 53BP1 and heterochromatin (please see more details below). Whilst the localization of 53BP1 at heterochromatin is convincing, the data can also support an independence between 53BP1-LLPS and HP1a-LLPS puncta. Further, dislodgement of 53BP1 from heterochromatin upon HP1a downregulation (Fig S12b) could simply be an indirect consequence of relaxation of heterochromatin, rather than a direct effect of 53BP1-HP1a interaction.

Response: *We have revised the co-IP data by repeating HP1 α , HP1 β and HP1 γ in Fig. S13a (now Fig. S10a after merging figures) and GFP in Fig. S13d (now Fig. S10d). We believe that these data now enhanced our conclusion that 53BP1 interacts with HP1 α , weakly with HP1 β , but not HP1 γ .*

While we understand the Reviewer's concern about the independence between 53BP1-LLPS and HP1 α -LLPS at heterochromatin, we feel that we have provided data to support the importance of the 53BP1-HP1 α interaction, but not heterochromatin relaxation, in 53BP1's LLPS formation at heterochromatin. For instance, when we depleted HP1 β or γ , we did not observe a defect in 53BP1 puncta/LLPS at heterochromatin as shown by HP1 α depletion (Fig. 3), which is consistent with the co-IP data. If heterochromatin relaxation is the underlying cause, then depletion of these two heterochromatin factors, at least HP1 β , should have produced a similar effect (i.e., reducing 53BP1 puncta at heterochromatin as a consequence of heterochromatin relaxation) as did by the depletion of HP1 α . Further, our co-IP data show that the interaction between 53BP1 and HP1 α depended on the OD domain, especially the four conserved residues in the OD of 53BP1, further supporting a specific effect of the 53BP1-HP1 α interaction. In combination with the immunofluorescence and ChIP-seq analysis, we hope that the Reviewer would agree with us that our data support the role of 53BP1-HP1 α interaction in the LLPS of 53BP1 at heterochromatin.

Additionally, the manuscript is largely requiring methodology details, rendering the ability to confirm some of the author's claims with the data presented. For example, "puncta rate" is widely used to characterize distinct 53BP1 foci and their function but there is no explanation on how 53BP1 foci/puncta are defined besides from a statement about the differences in number of foci (<3 or >4). The manuscript would be greatly improved by the addition of detailed information about methods, in particular imaging analysis, both in the text and in the figures to enable an appropriate interpretation of the data and experimental procedures by the reader.

Response: *We thank the Reviewer for raising this great point and apologize for not providing enough methodological details in the previous version of the manuscript. In this revision, we have carefully gone through the manuscript and provided detailed methods for our analysis about 53BP1 puncta (now we added puncta number and size data) and how we determined 53BP1 puncta vs DSB foci in the manuscript (Results, Methods and legend sections). We also provided details about*

imaging analysis in the Methods and figure legends. Please also see specific responses below.

Specific comments are detailed below.

Major comments:

1. Detailed description of methodology, in particular of immunofluorescence imaging analysis is lacking. “Puncta rate” definition has been included in lines 103-104 but information as to how the foci are defined (p.e. masking/thresholding manually or automatically, software used, etc) should be defined. Also, are the images shown projections or single-lanes? This information should be properly provided in the manuscript.

***Response:** We thank the Reviewer for raising this great point. We have added details about image analysis in the Method section (lines 730-753), as well as in figure legends.*

In brief, all images are single z-plane pictures captured by the Leica TCS SP8 confocal with HyVolution 2 except Fig. S9a (now Fig. S8c) that was projections. Images for each channel (i.e., blue, green or red) were exported as individual TIFF files by the built-in Leica LASX software and individually analyzed by Photoshop (version 21.2.12). Only Brightness/Contrast of the entire image was adjusted manually, but the values were kept the same for the same batch of images (i.e., those taken at the same day with the same parameters such as the laser intensity, filter, and gain values). The image size has always been set at 2048×2048 dpi for all confocal experiments. Image cropping and merging were also done in Photoshop. Scale bar was applied to individual image before cropping, which was also used to determine the puncta or DSB foci size.

2. Similarly, the number of foci/puncta per condition would be more informative than an arbitrary number to define puncta versus puncta. The authors should include histogram of densities to show the variation in number of 53BP1 (or H3K9me3) foci to strengthen a functional difference associated to 53BP1 foci number. This is crucial as most of the claims stem from the difference between puncta/foci and both previous reviewers and myself find the definitions provided quite insufficient.

***Response:** We thank the Reviewer for this great suggestion. We have re-analyzed nearly all quantitative data and generated histograms to show 53BP1 puncta density variation (specifically, distribution of puncta number per cell) and added them in Fig. 1d, 2f, 2h, 6c, 6f, 6i, 7a, S2c, S3c, S4c, S5c, S6c, S7c, S11d, S11g, S12c, S13d and S13f. Other figures like Fig. 2i, 3f, 3i, 4h, and 7e already showed 53BP1 puncta distribution in the previous version. We also added details about the analysis in the figure legends. Despite these changes, all the new data still supported our conclusions.*

3. The statistical information is also poorly described through the manuscript. Often the number of cells or foci analysed are not mentioned in the text, figures nor in the legends and when “n” is mentioned, there is often discrepancy to what data shows. For example, several figure legends mention that each dot represents a cell but the number of dots often does not match the “n” provided (for example S2B n= 425 for parental but there are only 37 dots).

***Response:** We apologize for not being clearer on these. In cases like Fig. S2b that showed **53BP1 puncta rate**, n=425 was the total cell number analyzed for that particular figure panel. However, when we previously calculated the puncta rate, we analyzed how many cells in a given image had*

53BP1 puncta. In this case, we analyzed 37 images taken from >three independent experiments. Hence, there were 37 dots, although the combined total cell number in these 37 images were 425. Nonetheless, we agree with the Reviewer that the description in the previous version was not clear enough and confusing.

*In the revision, we have now removed the labeling of cell numbers in all figure panels **that show puncta rate**. Instead, we described both the number of images and total cell number analyzed in the corresponding figure legend. But for other figure panels where the numbers match cells/events analyzed, we will display the numbers in the figure panels.*

4. Overall, the organization of the supplementary figures is very confusing. There are many supplementary figures for each main figure and a logic order following the mentions within the manuscript is not followed. For example, in the second paragraph of first result section (line 129-136), text refers to s8 and figure 7 (before s2-4 are even mentioned in the text). This back and forth organization is very distracting to the reader, further rendering proper interpretation of data. Strongly recommend authors to re-organize and merge the supplementary Figures.

Response: *We apologize for the confusion. We have carefully modified the manuscript to avoid this back and forth issue.*

As to the figure numbers, please note that the original Fig. S2-S7, Fig S8-S9 and Fig. S10-S11 were to show 53BP1 puncta in different cell lines. These data may sound redundant/repetitive, but are important to demonstrate the generality of 53BP1 puncta across various cell lines, which was also mentioned by the Reviewer as a strength of our study. Therefore, if we consider these as one 'big' supplementary figure (presenting the same concept), the real number of supplementary figures was not that many in our opinion.

Nonetheless, we have now merged Fig. S8-9 into new Fig. S8, Fig. S10-S11 into new Fig. 9 and Fig. S18-S19 into new Fig. S15. We also removed some redundant results, like the ones showing that the chemical 1,6-hexanediol inhibited puncta formation for 53BP1 and heterochromatin factors in the previous Fig. S15-S17. Despite the addition of more than 10 data panels from new experiments and analysis to address this Reviewer's critiques, we managed to reduce the number of supplementary figures from 19 to 15, which in essence should be 10, after merging and removing figure panels.

5. Often, the DAPI staining in the representative images selected for the parental example shows features of unhealthy cells (p.e. micronuclei or apoptotic-associated DAPI staining is visible in S3a, S9a, S9b, S10a, S10f). The authors should perform apoptosis analysis to demonstrate that the large 53BP1 foci numbers observed in their parental cell-lines are not resulting from stress during culturing conditions. This would further support that this puncta are spontaneous.

Response: We have answered a similar question raised by the previous Reviewer #2 during our revision to NCB, who asked if the dense DAPI staining in 53BP1-C1 expressing cells were caused by apoptosis. We showed that 53BP1-C1 puncta-expressing cells did not express any elevated level of cleaved Caspase, a marker of apoptosis (which was in Fig. S15a, but now in Fig. S12a), demonstrating that the condensed DAPI staining in 53BP1-C1 puncta-expressing cell was not due to cell death.

We understand that this Reviewer was concerned about cell culture conditions that may cause death of **parental** cells, which might have consequently induced the puncta. First, we have been very careful about cell culture conditions. Further, we only culture cells for about 10 passages, especially those 53BP1 KO cells. After that, we will thaw new vials for experiments.

Nonetheless, to answer this question by experiments, we examined cleaved Caspase 3 in 53BP1 puncta positive **parental** cells. Like what we observed for 53BP1-C1-expressing cells, we did not observe any increase in the level of cleaved Caspase 3 when 53BP1 formed puncta under normal growth conditions in parental cells. Yet, we could readily detect increases in cleaved Caspase 3 when we treated cells with the DNA damaging agent, bleomycin, which served as a positive control of cell death (in addition, there was no correlation between 53BP1 foci and the level of cleaved Caspase 3 in the presence of DNA damage). The results are shown here and also in Fig. S4e (lines 152-156). Please note that the arrow indicated a cell having abnormal nuclear shape as the Reviewer mentioned, but it did not show higher than normal level of cleaved Caspase 3. Hence, we believe that the puncta were not due to cell culture condition issues, but occurred spontaneously.

6. The claim that 53BP1 is required for heterochromatin formation is not substantiated by the data presented through the manuscript and therefore should be removed.

Response: We agree with the reviewer on this part. According to our data, what is more accurate is that 53BP1 is important for heterochromatin maintenance, but not formation. We have modified the manuscript accordingly.

7. Finally, authors should discuss if 53BP1 puncta inhibit repair-associated 53BP1 foci formation as puncta seem to occur concurrently with replication (cyclin A positive cells).

Response: This is a great point. As the Reviewer pointed out, since 53BP1 puncta were found in Cyclin A positive S-G2 phase cells where 53BP1 has been reported to be excluded or reduced from DSB sites by BRCA1, it suggests that formation of 53BP1 puncta could further inhibit foci formation of 53BP1 at DSB sites. Consequently, 53BP1 puncta could inhibit DSB repair by non-homologous end joining (NHEJ) while promoting DSB end resection. Hence, it would also be interesting to determine whether such 53BP1 puncta could then promote homologous recombination (HR)-based DSB repair. As we briefly discussed in the previous revision, we propose to address the impact of the LLPS function of 53BP1 on DSB repair at heterochromatin. We believe that we can also use similar approaches (specific DSB generation by CRISPR/Cas in conjugation with NHEJ or HR reporter

assays) to determine the role of 53BP1 puncta in DSB repair, which we hope to be conducted in a future study. We have added a discussion in the revision on this point (lines 582-588).

Minor comments:

9. Puncta size and intensity (of both 53BP1 and heterochromatin) are mentioned through the manuscript without any analysis of such features described or shown (p.e. lines 148-149, 216, 270-271). Please perform the analysis and include description of methods used both in legends and methods section.

Response: *As suggested by the Reviewer, we have performed analysis to determine the size of both the puncta and DSB foci of 53BP1, as well as heterochromatin centers in human cells (H3K9me3) and mouse cells (DAPI). We have added the data in Fig. S1g, S4g, 2h, S11f, and S14e. Description of how the puncta size was analyzed has been added in the Method (lines 748-753), the figure legends, and mentioned in the manuscript (lines 95-97, 166-169, 170-172, 295-297, and 469-472).*

For instance, for 53BP1 puncta formed by endogenous proteins, we determined their size to be $1.243 \mu\text{m}^2$ (median in area) with the 25% and 95% percentiles to be 0.680 and $5.178 \mu\text{m}^2$, respectively. Hence, we counted those 53BP1 dots that have an area greater than 0.680 as 53BP1 puncta (lines 95-97, 120-121). Similarly, we measured the size for the puncta formed by 53BP1-C1, and we found that the puncta size was bigger than endogenous 53BP1 (5.081 vs 1.243), which is consistent with the immunofluorescence images that showed larger puncta for 53BP1-C1. Interestingly, when we measured the size of DSB foci formed by 53BP1-C1, it turned out that the DSB foci were much smaller in size (median at $0.377 \mu\text{m}^2$) than the puncta (Fig. S14e), facilitated our discrimination of 53BP1 puncta from DSB foci.

As to the puncta intensity, we presented such kind of data in our very first submission to NCB, but was criticized by the previous Reviewer #2, whose opinion was that the puncta intensity depended on the fluorophore and experimental conditions and could vary with each experiment; therefore, Reviewer #2 commented that the puncta intensity was not a reliable index to determine the feature of puncta, nor to compare them among different groups. We agreed with Reviewer #2 on this point and removed those data during our revision to NCB and then to NC. Hence, we did not present the data of puncta intensity here. Nonetheless, the rate, the number and the size distribution of 53BP1 puncta are adequate to demonstrate the biological function in our opinion.

10. Figure 2: Related to reviewer #1 comment about 53BP1 location at heterochromatin, how were images acquired in different channels? 53BP1 staining seems to always border the left side of the heterochromatin-foci (H3K9me3 signal), which could be a consequence of plate-drift during imaging (if sequential acquisition was performed).

Response: *Images were indeed acquired sequentially under a confocal microscopy system. However, the Leica TCS SP8 confocal microscopy has an air cushion stabilizing system and has been routinely maintained by Leica, so there weren't any technical issues that could lead to plate drifting during sequential image capturing.*

As to the point raised by the Reviewer (53BP1 staining seemed to be always border the left side of heterochromatin), with all due respect, we do not think this is the case. Careful and close inspection showed that 53BP1 signal could be all around heterochromatin. The line scanning results in Fig. 2 can support this conclusion. What might have given the reviewer such an impression may be from

Fig. 2e, where we highlighted the co-localization between 53BP1 and H3K9me3 in 4 puncta. The first punctate from the left shows that the 53BP1 punctate is larger than that of H3K9me3 (but had more 53BP1 signal on the left side of H3K9me3); the second punctate was a complete co-localization; the third one is where 53BP1 signal was on the left side of H3K9me3; but the 4th one is the opposite where 53BP1 puncta were on the right side of H3K9me3. Also, if we look at the other 53BP1 punctum right beneath the 4th punctum in the same picture, we can see that the 53BP1 punctum was also at the right side of the H3K9me3 dot. Further, MEF cells clearly showed that 53BP1 wrapped around H3K9me3 in any directions like in Fig. 2h. Therefore, we hope that the Review would agree that there wasn't an issue of imaging capture.

11. Figure 2i: the y-axis defines #DAPI centers/cell but the interpretation of these results in the text mentions size differences, rather than number differences: "reduction in the number of large heterochromatin centers accompanied with an increase in smaller ones" (lines 148-149 and later in line 216). What is this panel showing? Please correct accordingly and add information on how analysis was performed, including how DAPI centers were defined.

Response: We apologize for the unclearness. Previous Fig. 2i showed the number of DAPI-indicated heterochromatin centers in parental and 53BP1 KO MEF, but apparently, we should have also included data to show the size reduction. As mentioned above (point #9), we now summarized the heterochromatin center size for human cells in Fig. S4f and for MEFs in Fig. 2h (lines 166-168, 170-172), which showed a significant reduction in the size of heterochromatin centers in 53BP1 KO cells compared with parental cells, supporting our conclusion.

12. Figure 2f&g: There are 40-50% of cells in the parental line with less than 3 H3K9me3 foci. This result is quite surprising as H3K9me3-foci are quite prevalent in all cell-types. Details on how H3K9me3-foci were defined need to be provided to enable the interpretation of an otherwise unexpected result.

Response: With all due respect, we think parental cells in Fig. 2b, 2e and 2h (previous Fig. 2f was for dot line scan and Fig. 2g was a summary of H3K9me3 puncta rate; hence, we believe that the Figure 2f & g was a mis-labeling as these two panels were not meant to show H3K9me3 puncta in cells) had multiple H3K9me3 foci (at least >3), which in our opinion matched with previous publications. We speculate that maybe the Reviewer was mentioning Fig. 3 or Fig. S10-S11 where we showed HP1 $\alpha/\beta/\gamma$ -defined heterochromatin centers in human cells? To the best of our knowledge, it is known that HP1 proteins do not form clear puncta like H3K9me3 in human cells.

In the meantime, we do agree with the Reviewer that we have not clearly defined how we quantified H3K9me3 puncta rate in these figures. In the revision, we have now measured the H3K9me3 puncta size in human cells and showed that the median size of H3K9me3 puncta was significantly reduced from 1.152 μm^2 in parental cells to 0.682 μm^2 in 53BP1 KO cells (Fig. S4g) (lines 166-168). Similarly, when we measured the mouse heterochromatin centers indicated by the bright DAPI staining, we noticed a reduction in the size of heterochromatin centers from 1.450 in parental cells to 0.408 μm^2 in KO cells (Fig. 2h) (lines 170-172). The sizes of the 25%, 75% and 95% percentiles of heterochromatin centers in human and mouse cells were all reduced in 53BP1 KO cells (Fig. S4f and Fig. 2h), supporting our conclusion that depletion of 53BP1 led to a reduction in the size of heterochromatin centers.

Using the top 25% percentile as a cut-off to determine the large heterochromatin centers in human cells, we found that 53BP1 KO reduced the number of these large bright heterochromatin centers (Fig. 2f), which is the same as in MEF cells. Since heterochromatin centers are readily visualized in MEFs, we found that the reduction in the number of large heterochromatin centers was accompanied with an increase in the total number of heterochromatin centers (large and small combined) in 53BP1 KO MEFs (Fig. 2i). These results together support the idea that 53BP1 is important for maintaining the structure of heterochromatin, especially the large heterochromatin centers.

13. Figure 4: Please define mcBox, SatIII and ALR to allow for a better understanding of the heterochromatic loci probed, similarly to the description provided to murine major and minor satellites.

Response: *We thank the Reviewer for pointing this out. mcBox, SATa (also called ALR, so we have unified the labeling in Fig 4 to be SATa), SATIII, and CH21 (later on in point # 24) are all AT-rich alpha satellite repeats located on centromeric chromosomes in primate, an array of which could extend to mega-bases in length and forms heterochromatin structure (Nishibuchi et al, 2017 and Francastel et al, 2019). We have added this description in the revised manuscript (lines 247-249, 268-270 and 546-547).*

14. Figure S15a does not show that “53BP1-C1 puncta completely overlapped with HP1a or H3K9me3”, as stated in line 251. There is a better overlap between marks but many HP1a or H3k9me3 foci are not overlapping with 53BP1-C1. Please correct statement.

Response: *We thank the Reviewer very much for correcting us on this. We have changed the manuscript accordingly by saying that the percentage of 53BP1-C1 puncta that co-localized with heterochromatin was significantly higher than 53BP1-FL. Further, for the co-localized puncta, 53BP1-C1 demonstrated nearly complete co-localization, whereas 53BP1-FL mainly wraps around or abuts heterochromatin centers, displaying as partial co-localization (lines 298-302).*

15. Line 252-253: “These results suggest that 53BP1-C1 promoted heterochromatin formation”. These results do not support this claim, but rather suggest that 53BP1 is recruited to heterochromatin. Please remove this phrase.

Response: *We thank the Reviewer for pointing this out and have removed this sentence. Instead, we stated as suggested by the Reviewer (lines 302-303).*

16. Figure S16a: The images chosen are oversaturated and the claim that puncta can be observed through the interphase and are resolved during mitosis, as claimed in the manuscript (line 275-276), cannot be confirmed. Please provide better images to support this statement. Also, times are typically shown as timepoints (t0, t2, t4, t10, etc) as the actual timing of acquisition is not informative.

Response: *We have reduced the image intensity so that they will not be oversaturated, which we believe is better to show the puncta change during cell cycle, which is now in Fig. S13a.*

We also thank the Reviewer for the suggestion to change the acquisition time to sequential time points.

17. Please explain the score in S16b in the legend.

Response: *The IUPred2 software is to identify Intrinsically Disordered Protein Regions (IDPRs, i.e. regions that lack a stable monomeric structure under native conditions). When inputting any given peptide sequences, the program returns a score between 0 and 1 for each residue, corresponding to the probability of the given residue being part of a disordered region. The graph shows the disorder tendency of each residue in human 53BP1, where higher values correspond to a higher probability of disorder. We have added this in the figure legend, which is now Fig. S13b.*

18. For the 1,6-hexanediol treatment, why was it performed in fixed cells? Live imaging with the constructs GFP-53BP1-C1 and/or GFP-HP1a shown by the authors would be more adequate to show that these foci are sensitive to this alcohol. Further, as the authors use mCherry-HP1a, these experiments in cells co-expressing both constructs would additionally show whether 53BP1 and HP1a belong to the same LLPS foci or not.

Response: *We wanted to observe the effect of this chemical on puncta formation by endogenous 53BP1 and heterochromatin proteins, so we used fixed cells in the original Fig. S16-S17.*

Nonetheless, as suggested by the Reviewer, we now performed live cell imaging experiments to show that 1,6-hexanediol reduced the puncta number formed by GFP-53BP1-C1, as well as the ones co-localized by GFP-53BP1-C1 and mCherry-HP1 α , which is in Fig. S13e-f (lines 336-338).

Also, to address the Reviewer's concern about too many supplementary data, we have removed the chemical treatment data for endogenous HP1 α and SUV39H1 in the previous Fig. S17a-c, which showed the same effect of 1,6-hexanediol on reducing the puncta formed by GFP-53BP1-C1, HP1 α or SUV39H1. Since the newly added live cell imaging data (Fig. S13e-f) showed the same results as the previously fixed cell data, we believe that removal of these data does not affect our conclusion.

19. Fig17a-c: these images are oversaturated making it difficult to interpret the same as the authors did. In particular, 53BP1-C1 doesn't seem to be affected by 1,6-hexanediol treatment, as puncta are still visible (particularly in S17c).

Response: *We believe that the Reviewer was mentioning images for DAPI in Fig. S17a and SUV39H1 and DAPI in Fig S17c in DMSO treated control cells, whose brightness was increased with a purpose to visualize the puncta in other cells in these images, which unfortunately made some cells looking like oversaturated. However, our conclusion about the reduction in 53BP1-C1 or heterochromatin puncta by this chemical was not affected.*

In the revision, we have now removed the results in previous Fig. S16c-S17d. Currently, we only show how 1,6-hexanediol treatment reduced the puncta formation of endogenous 53BP1 in fixed cells (Fig. S13c-d) and GFP-53BP1-C1 and mCherry-HP1 α in living cells (Fig. S13e-f). These data support our previous conclusion that 1,6-hexanediol treatment reduced puncta formation of 53BP1 and HP1 α . Hence, we believe that removing the previous Fig. S17a-c data does not change our conclusion.

As to the reviewer's point that 53BP1-C1 puncta were not affected by 1,6-hexanediol, in our humble opinion, this chemical did reduce both the number and the frequency of 53BP1-C1 puncta, supporting the idea that these 53BP1 puncta represented liquid-liquid phase separation. From both

our experience and published literatures, 1,6-hexanediol treatment cannot completely abolish LLPS puncta in cell cultures, including those formed by HP1 α (Chong S et al, 2018). So our observations are consistent with published results.

Yet, the significant reduction in the puncta number formed by 53BP1 or heterochromatin factors supports the idea that these puncta are LLPS. We have modified our conclusion in the corresponding section by saying that 1,6-hexanediol significantly, although not completely, reduced 53BP1-C1 puncta formation (lines 335-336).

20. Figure S17d: Do cells survive after 5-10min of 1,6-hexanediol treatment? Most studies published in different cells (and organisms) show that this alcohol is toxic after 10min, leading to apoptosis so this result is very surprising. Please discuss.

Response: We are well aware of reports saying that 1,6-hexanediol is toxic, especially when used at 10% of concentration. Hence, we used a concentration of 5%, which was much less toxic. For that particular experiment in the previous Fig. S17c, we were able to acquire data from U2OS cells after 1,6-hexanediol treatment for up to 120 min. Nonetheless, we agree with the Reviewer that such a treatment may have caused cell death, which may compromise the data interpretation.

Also, by combining with points mentioned above (to reduce the number of supplementary figures, and to use GFP-53BP1-C1 and mCherry-HP1 α to examine the effect of this chemical on puncta formation in live cells), we think it is best to remove this piece of data. Since we have shown in the new Fig. S13e-f about the reduction in puncta formation for GFP-53BP1-C1 and mCherry-HP1 α by this chemical in live cells, it is not necessary to show the time-course quantitation again. Importantly, removing this piece of data does not affect our conclusion.

21. Figure 5c: Why are the droplets formed hollow in the center? LLPS droplets are formed by separation of interacting molecules from the solution, which creates a homogeneously isolated foci. Thus, the empty core of the droplets shown needs to be discussed. Also, an experiment where increased concentrations of either HP1 α or 53BP1-truncations would probe for their dependency in forming LLPS droplets.

Response: This is a great point that we only briefly discussed in the previous submission due to manuscript length limitation. While we agree with the Reviewer that classically, liquid droplets are considered to be solid condensates, there are a number of studies showing hollow sphere-like liquid droplets, for instance, those formed by germ granules of *Drosophila* (Arkov et al, 2006; Kistler et al, 2018), transcription factor TDP-43 (Chmidt et al, 2016), and RNA-RNPs (Banerjee et al, 2017; Alshareedah et al, 2020). Such vesicle-like liquid droplets were considered to be formed by anisotropic protein-protein or protein-nuclei acid complexes that are probably driven by heterotypic electrostatic interactions between molecules. Hence, our observation of the hollow structures of 53BP1 and HP1 α is not something that is totally unexpected, and we assume that they might also be formed in a similar way.

In our opinion, there are a couple of reasons that could contribute to the specific liquid droplets formed by 53BP1 and HP1 α . First, such kind of vesicle-like liquid droplet formation might be a unique feature for 53BP1 and HP1 α at the particular concentrations used. Of note, the ones with hollow center are often the liquid droplets that were relatively large in size (with a diameter up to 20 μm) and with higher component concentration, which is consistent with the anisotropic interactions

seen in RNA-RNPs (Alshareedah et al, 2020). When we looked to the smaller droplets, especially at lower concentrations of 53BP1 and HP1 α , we did see that the majority of the condensates were the widely reported solid droplets as shown in Fig. 5c and the new Fig. S14c-d. Second, there is another possibility that if nucleosomal components were included, they might form conventional solid droplets, which will be explored in detail in a future study. We have added a discussion in the revision on this observation (lines 613-629).

Further, we carried out additional experiments to show that increasing the concentration of 53BP1-C1 enhanced the liquid droplet formation of HP1 α , and vice versa (new Fig. S14c-d) (lines 351-353), addressing the reviewer's concern and also further supporting our conclusions.

22. Figure 5f: the images in HP1a KD panel at 60 and 120 seconds are confusing ... At 60 seconds the bottom highlighted chromocenter shows recovery of fluorescence but it is again lost at 120 seconds?

Response: We applaud the Reviewer for careful inspection of the data. First, these were the real data, which were not caused by any image manipulation. Second, in our opinion, this transient appearance and then disappearance of 53BP1 puncta signal at heterochromatin in HP1 α depleted cells perfectly reflect the importance of HP1 α in the stable recruitment of 53BP1 to heterochromatin, supporting our conclusion. We have added a discussion in the revision on this part (lines 374-375, 389-391).

23. Figure 6: The number of cells analysed has not been included in the Figure or legend.

Response: We thank the Reviewer for pointing this error out. We have included cell numbers analyzed either in the figure panels or in the figure legend.

24. Figure 8f: Please define CH21 (as in comment 13 above).

Response: We have defined CH21 in point #13, which is also an alpha satellite repeat at human heterochromatin (lines 546-547).

25. Figure S13: ChIPseq analysis is very interesting but a genome wide analysis should be shown (for example showing the number of overlapping peaks). The 5 loci presented are not enough to claim that the overlap observed by immunofluorescence is corroborated by ChIPseq data.

Response: As suggested by the Reviewer, we analyzed genome wide association between 53BP1 and H3K9me3 using publicly available data from U2OS cells. Peaks of 53BP1 and H3K9me3 intersected with γ H2AX were filtered to avoid the effect of DNA damage on the co-localization. The analysis shows that about ~23% of 53BP1 signals overlapped with H3K9me3, indicating a preferential binding of 53BP1 to heterochromatin. The results of a Heat map have been included in the new Fig. S10b in the manuscript (lines 202-204), which supports our conclusion.

26. Line 371-374: The blot presented to show that OD-4A interacts less with FLAG-HP1a is not very robust. The authors should replace for a more clearer (stronger GFP band), which would better allow to infer a lower interaction.

Response: *We have repeated this co-IP experiment (also mentioned at the main points) and presented the new result, which shows clearer GFP blot as shown in the new Fig. S10j. These data support our conclusion that the conserved residues in the OD domain are important for the 53BP1-HP1 α interaction.*

27. Line 390: The different definitions of the foci according to treatments seems very subjective. Again, a simple histogram density distribution of number of foci (automatedly defined) should unbiasedly show differences in foci formation.

Response: *As described above in point #2, as well as from suggestions from Reviewer #1, we now used co-localization between 53BP1 (FL or mutants) and one of the three DSB repair markers, γ H2AX, pKAP1 or RIF foci, as a more reliable readout to discriminate between 53BP1 puncta/LLPS and DSB foci. From our extensive analysis, we found that when 53BP1 formed DSB foci, they largely co-localize with foci formed by one of these three DSB markers. However, when 53BP1 formed the puncta, there will be no to limited co-localization between 53BP1 puncta and the DSB marker foci (see example in Fig. S12b). Hence, we now use these criteria to distinguish between 53BP1 puncta and DSB foci.*

In addition, we have presented histogram data to demonstrate the size distribution for 53BP1 puncta and DSB foci analyzed by the Image J software, which shows that the median size of 53BP1 puncta is significantly larger than the DSB foci, which also helped to define 53BP1 puncta over DSB foci (lines 312-324, 418-431, 466-470).

28. Line 481: “reserve” should be “preserve”.

Response: *We thank the Reviewer for pointing this error out. We have changed it in the revision (now line 571).*

29. Axis of puncta rate should be standardized to show 0-100, for consistency.

Response: *We have changed all the y-axis of figures presenting puncta rate into 0-1.0 for consistency, which include Fig. 1c, 3c, 3h, S2b, S3b, S4b, S5b, S6b, S7b. In other figures, we have used puncta distribution graph.*

30. Please replace “heterochromatins” with “heterochromatin”.

Response: *We have changed it in the revision.*

31. Line 168: “binging” should be “binding”.

Response: *We thank the Reviewer for pointing this error out. We have changed it (now line 196).*

32. Line 227: please change phrase to “53BP1-FL/WT slightly reduced, although significantly, the elevated levels of heterochromatin” to better reflect data shown.

Response: *We have changed it as suggested by the Reviewer (now lines 271-272).*

33. Line 249: Please change to “ 53BP1-C1 puncta elicited an increased number of DAPI foci” to more accurately reflect the data.

Response: *We have changed it to “cells expressing 53BP1-C1 puncta also had increased number of DAPI puncta” (now lines 292-293).*

REVIEWERS' COMMENTS

Reviewer #1 (Remarks to the Author):

Over the course of several rounds of revision, the authors have invested extensive efforts to address my concerns and those of the other referees. In the new version of the manuscript my main outstanding concerns have now been satisfactorily addressed, and I therefore find the study acceptable overall for publication. However, I would recommend that the manuscript text be carefully edited by a native English speaker before formal acceptance and publication of the paper.

Reviewer #4 (Remarks to the Author):

The authors have satisfactorily addressed my major concerns, in particular the lack of methodology related to puncta definition and of microscopy and statistical analysis. The authors have also opted to remove any data and consequently text related to a putative role of 53BP1 in heterochromatin function (with few exceptions, please see below), which I support.

While I still think the manuscript largely lacking in mechanistic insights into the novel role of 53BP1 in heterochromatin, the authors have performed an extensive characterization of this phenomena across cell-lines and which 53BP1 domains are responsible for the separation of function at heterochromatin and DSB repair. Thus, I find the current manuscript overall acceptable for publication in Nature Communications, with the following small changes:

1. Page 2, line 9: please correct this phrase in the abstract to: "rescued heterochromatin -de-repression and protected cells from stress-induced DNA damage and senescence".
2. Page, line 125: only Fig 1e shows cyclin A staining, please remove reference to Fig. 1b.
3. Page 12, line 259: Please remove "function" from the title of the result section. As mentioned previously, the current manuscript shows no data supporting a role, much less a requirement, for 53BP1 in maintaining heterochromatin function.

Manuscript number: #NCOMMS-21-33442A

- We thank Reviewer #1 and Reviewer #4 for the positive comments and constructive critiques. We have revised the manuscript to address these issues.
- Changes are highlighted in blue in the revision.

REVIEWER COMMENTS

Reviewer #1 (Remarks to the Author):

Over the course of several rounds of revision, the authors have invested extensive efforts to address my concerns and those of the other referees. In the new version of the manuscript my main outstanding concerns have now been satisfactorily addressed, and I therefore find the study acceptable overall for publication. However, I would recommend that the manuscript text be carefully edited by a native English speaker before formal acceptance and publication of the paper.

***Response:** We appreciate the reviewer for his/her constructive comments during the review process, which significantly improved the manuscript. We also thank the reviewer for the positive comment on this revision and the support of its publication at Nature Communications.*

I have asked my colleague, Dr. John Pink, an Assistant Professor and the Director of the Translational Research Shared Resource at Case Cancer Center, to read the revised manuscript. Dr. Pink is a native speaker, has sufficient knowledge on genome stability and DNA damage, and has helped us on other cases previously. Therefore, we believe that Dr. Pink is qualified to read our content and edit the language and grammars appropriately. Edited parts are highlighted in blue in the revised manuscript.

Reviewer #4 (Remarks to the Author):

The authors have satisfactorily addressed my major concerns, in particular the lack of methodology related to puncta definition and of microscopy and statistical analysis. The authors have also opted to remove any data and consequently text related to a putative role of 53BP1 in heterochromatin function (with few exceptions, please see below), which I support.

While I still think the manuscript largely lacking in mechanistic insights into the novel role of 53BP1 in heterochromatin, the authors have performed an extensive characterization of this phenomena across cell-lines and which 53BP1 domains are responsible for the separation of function at heterochromatin and DSB repair. Thus, I find the current manuscript overall acceptable for publication in Nature Communications, with the following small changes:

***Response:** We thank the reviewer for positively commenting on our effort to revise the manuscript and the support for the publication of our manuscript at Nature communications.*

1. Page 2, line 9: please correct this phrase in the abstract to: “rescued heterochromatin -de-repression and protected cells from stress-induced DNA damage and senescence”.

Response: We thank the reviewer for this suggestion, which has been changed in the manuscript in the Abstract (line 37-38).

2. Page, line 125: only Fig 1e shows cyclin A staining, please remove reference to Fig. 1b.

Response: We thank the reviewer for pointing this out. We have removed the reference of Fig. 1b here, which is now in line 110 due to the removal of embedded figures and reformatting to fit the journal style.

3. Page 12, line 259: Please remove “function” from the title of the result section. As mentioned previously, the current manuscript shows no data supporting a role, much less a requirement, for 53BP1 in maintaining heterochromatin function.

Response: We have removed the word ‘function’ in this heading, which is now in line 210 after editing the manuscript to fit the journal style.